

# Spring phenology and phenology-climate links inferred from two remotely sensed vegetation indices across regions and biomes

Xiyan Xu[1], William J. Riley[2], Charles D. Koven[2], Gensuo Jia[1*]

[1]Key Laboratory of Regional Climate-Environment for Temperate East Asia, Institute of Atmospheric Physics, Chinese Academy of Sciences, Beijing 100029, China

[2]Climate and Ecosystem Sciences Division, Lawrence Berkeley National Laboratory, Berkeley, California, USA

*Corresponding Author:

Dr. Gensuo Jia

Key Laboratory of Regional Climate-Environment for Temperate East Asia,

Institute of Atmospheric Physics, Chinese Academy of Sciences,

Beijing 100029, China

Email address: jiong@tea.ac.cn

Phone number: 86-10-82995314



**Abstract**

2       The timing of spring greenup (SG) as inferred by remotely sensed vegetation

indices have showed contrasting dynamics across the same region and periods.
Assessing the uncertainty in SG associated with different Normalized Difference
Vegetation Index (NDVI) products is essential for robustly interpreting the links
between climate and phenological dynamics. We compare SG inferred from two
NDVI products over the period 2001-2013: (1) Terra Moderate Resolution Imaging
Spectroradiometer (MODIS) and (2) National Oceanic and Atmospheric
Administration's (NOAA's) Advanced Very High Resolution Radiometer (AVHRR)
instruments processed by the Global Inventory Monitoring and Modeling Studies
(GIMMS) to explore confidence and uncertainty in the NDVI-inferred SG trend and
its links to climate variability. Both MODIS and GIMMS agreed in showing an
advancement of SG in northern Canada, the eastern United States, and Russia, as well
as a delay in SG in western North America, parts of Baltic Europe and East Asia. In
the regions with advanced SG, GIMMS inferred much weaker advancement whereas
in the regions with delayed SG, GIMMS inferred much stronger delay than MODIS.
This resulted in a GIMMS SG delay in both North America and Eurasia. MODIS data
show no significant SG shift in North American for spatial heterogeneity in SG shift,
but dominant SG advancement in Eurasia. The SG advancement inferred from
MODIS is associated with a stronger coupling between SG and temperature and a
stronger sensitivity across biomes as compared to GIMMS. The main uncertainty in
the SG trend and SG-temperature sensitivity are in northern high latitudes (>50°N)
where GIMMS and MODIS show different magnitude and sign of the annual SG
anomalies. Compared to 1988-2000, inter-biome GIMMS SG-temperature sensitivity





is stable and the SG-temperature sensitivity increased in the boreal and Arctic biomes
despite a slight reduction in the SG-temperature coupling over the period 2001-2013.
The explanation for the increased SG-temperature sensitivity remains unclear and
requires further investigation. We suggest broader evaluation of the NDVI products
against field measurements and inter-validation for robust assessment of vegetation
dynamics.
**Keywords**: NDVI, MODIS, GIMMS, phenology, spring greenup, sensitivity





## 1. Introduction

Vegetation phenology plays an important role in regulating land-atmosphere
energy, water, and trace-gas exchanges. As the time spanned by satellite-based
Normalized Difference Vegetation Index (NDVI) products has increased to longer
periods, several studies have used NDVI to derive spring greenup time (SG) at
regional and global scales. Several changes in SG have been documented in the past
half-century in response to ongoing climate change. The Northern Hemisphere SG has
advanced in the range of 0-12 days per decade as inferred by NDVI (Table 1). The
wide range of SG shifts stem from studies covering different periods and regions, and
different methods and datasets that have been applied to derive phenology metrics.
Many factors associated with the obtaining of satellite data—e. g. drift of
satellite orbits, calibration uncertainties, inter-satellite sensor differences, bidirectional
and atmospheric effects—may cause uncertainties in satellite derived data time series
and thereby the uncertainties in interpreting the vegetation dynamics. Four NDVI
products have been published based on radiances collected by the Advanced Very
High Resolution Radiometer (AVHRR) instruments carried by programs of
NOAA/NASA Pathfinder  (PAL): Global Inventory Monitoring and Modeling Studies
(GIMMS), Land Long Term Data Record (LTDR) version 3 (V3) and Fourier-
Adjustment, Solar zenith angle corrected, Interpolated Reconstructed (FASIR). Each
of these records extends back to the year 1981. Because of their long time span, the
AVHRR NDVI products have been applied in numerous regional to global vegetation
phenology studies (Table 1). Advantages are recognized for GIMMS NDVI over the
other AVHRR NDVI products to represent the temporal variation of NDVI (Beck et
al., 2011). The more recent NDVI products retrieved from Terra Moderate Resolution
Imaging Spectroradiometer (MODIS) and Système Pour l'Observation de la Terre





(SPOT) VEGETATION mission (1 km)(e.g., Durpaire et al., 1995) are considered an
improvement over AVHRR for improved calibration and atmospheric corrections, and
higher spatial resolution (Zhang et al., 2003).
Several inter-comparisons have been conducted to evaluate the quality of
different NDVI products. Yet broad validation of NDVI products by using field
measurements is limited. The SPOT-4 VGT was used to evaluate the AVHRR PAL
(1998-2000) and AVHRR GIMMS (1998-2004) NDVI time series for African
continent. The dynamic range of SPOT-4 VGT NDVI is generally higher than the
AVHRR PAL NDVI, but matched GIMMS NDVI, implying an improvement of
GIMMS over PAL (Fensholt et al., 2006), however, the growing season GIMMS
NDVI is lower than MODIS NDVI in African semi-arid environment (Fensholt and
Sandholt, 2005). The annual average trend of GIMMS NDVI is consistent with
MODIS NDVI in the semi-arid Sahel zone, but higher discrepancies in the more
humid regions (Fensholt et al., 2009).  In the north 50°N, four NDVI products
(GIMMS3g, GIMMSg, SeaWiFS, SPOT) except MODIS showed consistent greening
trend over overlapping period although differences in growing season NDVI and
magnitude of greening trend pose uncertainties in satellite vegetation dynamics (Guay
et al., 2014). In mixed grassland in the Grasslands National Park of Canada, both
MODIS and AVHRR NDVI cannot quantify the spatial variation in ground based leaf
area index measurements (Tong and He, 2013).
Despite inconsistencies and uncertainties among these NDVI products,
GIMMS NDVI has been combined with other NDVI products to explore a longer
period vegetation dynamics or to constrain potential data quality issue. Zhang et al.
(2013) merged GIMMS NDVI over 1982-2000 with SPOT-VGT NDVI over 2001-
2011 to investigate the SG in the Tibetan Plateau. GIMMS SG over 2001-2006 was



discarded for its delayed SG trend, in contrast to SPOT-VGT and MODIS SG trend,
which was considered as a potential GIMMS NDVI data quality issue in the western
Plateau. SG trend in Tibetan Plateau advanced by about 10.4 days decade$^{-1}$ over 2001-
2012 inferred from merged GIMMS and SPOT-VGT NDVI (Zhang et al., 2013), in
contrast to the insignificant SG trend over 2000-2011 inferred from single GIMMS
NDVI (Ding et al., 2016).  The differences between GIMMS SG and SPOT-VGT and
MODIS SG were also found after 2000s in western Arctic Russia where values and
trends of MODIS and SPOT-VGT SG agreed very well (Zeng et al., 2013).  When
GIMMS NDVI was stitched with MODIS NDVI, the advancing trend of spring
greenup in Northern Hemisphere over 2002-2012 that was inferred from MODIS
NDVI is almost 3 times larger than the trend over the period 1982-2002 inferred using
the GIMMS NDVI (Wang et al., 2016). However, a similar study using the GIMMS
NDVI time series over 1982-2008 revealed an insignificant advancing trend in
Northern Hemisphere over 2000-2008 in relative to 1980-1999 (Jeong et al., 2011).
As the different methods in determining SG may not lead to such a high difference in
SG trend (Cong et al., 2013), we hypothesize the different NDVI products may lead to
the contradictory SG trend.

In this study, we attempt to better understand the causes of the differing findings

of SG trend in previous studies. We compared SG as inferred by GIMMS and MODIS
NDVI and their respective sensitivities to climate over the period 2000-2013, in
which both the AVHRR and MODIS instruments were active. We used an
independent climate reanalysis dataset to analyze the preseason, the period preceding
SG during which the climate drivers regulate SG, and the sensitivity between
preseason climate and SG. Data and methods are described in section 2. The results of
comparison of GIMMS and MODIS SG, the preseason climate that regulates the SG





and sensitivities of the SG to preseason climate are presented in section 3. Discussion
and conclusions are given in section 4 and 5, respectively.
**2.  Data and Method**
**2.1 Study area and biomes**

We restricted our analysis to north of 30°N, where temperate and boreal

vegetation dominate, since that is the region where phenology is expected to be most
strongly controlled by the annual cycle of temperature and moisture availability. In
order to analyze the phenology and its response to climate across biomes, we used
global mosaics of collection 6 MODIS data products (MCD12Q1) in the IGBP
classification of land cover types with spatial resolution of 0.5° x 0.5° to mask the
satellite-based SG results. The global mosaics of MCD12Q1 with geographic
coordinates of latitude and longitude on the WGS 1984 coordinate reference system
(EPSG: 4326) (Channan et al., 2014) were re-projected from standard MCD12Q1
with 500m resolutions (Friedl et al., 2010). We used the IGBP land cover
classification for 9 biomes in 2012 (Table 1): Evergreen Needleleaf Forest (ENF),
Deciduous Needleleaf Forest (DNF), Deciduous Broadleaf forest (DBF), Mixed
Forest (MF), Open Shrublands (OS), Woody Savannas (WS), Grassland (GL),
Permanent Wetland (PW), and Cropland (CP). We distinguish the grassland to the
north of 60°N (GLN), which is more likely to be tundra, from grassland in the
temperate south (GLS) due to their expected differences in climate and controls on
phenology.


**2.2 Climate reanalysis**

We calculated daily mean air temperature ($T_m$) and cumulative precipitation

($P_c$) from 6-hourly, half-degree resolution CRU-NCEP (Climate Research Unit-
National Centers for Environmental Prediction) v6 reanalysis to identify the preseason
climate associated with SG. The CRU-NCEP v6 dataset, recently extended to 2014, is
a combination of CRU TS v3.2 0.5° x 0.5° monthly climatology and NCEP reanalysis
2.5° x 2.5° with six hours time step available in near real time
(http://forge.ipsl.jussieu.fr/orchidee/wiki/Documentation/Forcings).
**2.3 NDVI products**

We used the latest version NDVI time series (GIMMS NDVI3g) derived from

the AVHRR instrument on board the NOAA satellite series. This dataset spans the
period from July 1981 to December 2013 with spatial resolution of 1/12° and
bimonthly temporal resolution (Pinzon and Tucker, 2014).

We also used the 16-day MODIS NDVI composites (MOD13C1, collection 6)

at 0.05° spatial resolution, and further performed data quality control. We regridded
both GIMMS and MODIS NDVI data to 0.5° x 0.5° resolution by taking the mean
value in a 0.5° x 0.5° pixel to match the spatial resolution of the CRU-NCEP
reanalysis. For GIMMS NDVI3g, the algorithm has improved snow-melt detection
and the pixels recognized with snow or ice were filled with average seasonal profile
or spline interpolation (Pinzon and Tucker, 2014). The pixels flagged with snow/ice
were given the NDVI values with the values from the previous nearest period without
snow influence. Even though, the filled values are very close to zero in the dormant
season and the near-zero values are smoothed by the piecewise logistic method
described in section 2.3. SGs were derived from GIMMS NDVI 2001-2013 to fit the



time period of MOD13C1.
**2.4 Determination of SG and preseason climate**
We determined the preseason duration following the method of Shen et al.
(2014), but with a different climate reanalysis product and method for calculating SG.
We restrict our analysis to north of 30°N, where temperate and boreal vegetation
dominate, since that is the region where phenology is expected to be most strongly
controlled by the annual cycle.
**Day SG and mean day of SG**
We first applied piecewise logistic method (Zhang et al., 2003) to fit and
smooth the temporal variation of vegetation index data (NDVI) to vegetation growth:
$$y(t) = \frac{c}{1+e^{a+bt}} + d \qquad\qquad (1)$$
where $t$ is time in days, $y(t)$ is the vegetation index at time $t$, $a$ and $b$ are fitting
parameters, $c+d$ is the maximum vegetation index value,  and $d$ is the initial
background vegetation index, usually the minimum vegetation index value preceding
the growing season. $D_{SG}$ is identified as the Julian date at which the rate of change in
the vegetation growth ($y(t)$) is maximum. $D_{SG}$ is the maximum of the curvature and
derived as the second derivative of equation (1) . The mean $D_{SG}$ ($\overline{D}_{SG}$) in each pixel is
averaged over the analysis years. For the pixels with multiple growth cycles in a year,
we applied this piecewise logistic method to the first cycle, so that $D_{SG}$ is the Julian
date at which the second derivative of $y(t)$ is maximum for the first time in a year.
**Preseason period and preseason climate**
We calculated the preseason period separately for temperature and
precipitation. To do this, we first calculated $T_m$ and $P_c$ during the respective preseason



periods. We defined the preseason climate ($T_\mathrm{m}$ and $P_\mathrm{c}$) in each pixel over the period
preceding $\overline{D}_{SG}$ from 15 to 120 days with an increment of 3 days. We expect the
relative variation in precipitation to be more relevant than absolute values in
determining phenology, thus we used the relative variation of cumulative precipitation
in percentage (%) of precipitation change instead of the absolute cumulative
precipitation variation in millimeter (mm). We detrended the calculated $T_\mathrm{m}$ and $P_\mathrm{c}$
over the historical period. For each period preceding $\overline{D}_{SG}$ for a given pixel, we
calculated the Pearson's correlation coefficients (PCC) between $D_{SG}$ and $T_\mathrm{m}$ (and $P_\mathrm{c}$).
We screened the data to remove pixels where we found a positive interannual
correlation between (1) preseason temperature and $D_{SG}$ and (2) preseason
precipitation and $D_{SG}$, respectively. We defined the period with the most negative
correlation between $D_{SG}$ and $T_\mathrm{m}$ (and $P_\mathrm{c}$) as the preseason $P_T$ (and $P_P$). The length of
preseason (days) for temperature and precipitation control is defined as $L_{PT}$ and $L_{PP}$,
respectively. The superscript of $G$ and $M$ represents the variables derived from
GIMMS and MODIS, respectively (e.g. $D_{SG}^M$ and $L_{PT}^M$ are $D_{SG}$ and $L_{PT}$ derived from
MODIS, respectively.).

**SG response to preseason climate**

We calculated the response of SG to preseason climate by calculating linear

regressions between $D_{SG}$ and $T_\mathrm{m}$ (and $P_\mathrm{c}$). We excluded the $SG$ response to preseason
climate in pixels where no significant relationship was found (i.e., $p$-value $> 0.1$).
**3. Results**
**3.1 MODIS and GIMMS SG comparison**

The spatial pattern of GIMMS-inferred mean $D_{SG}$ ($\overline{D}_{SG}^G$) and MODIS-inferred

$D_{SG}$ ($\overline{D}_{SG}^M$) is consistent ($r = 0.83$, $p < 0.01$). The regions with evident difference





between $D_{SG}^G$ and $D_{SG}^M$ are in the circumpolar Arctic and Asia high-altitudes (Figure 1a
and 1b) where correlations between the time series of $D_{SG}^G$ and $D_{SG}^M$ are relatively low
(Figure S1a). About 47% of the pixels in the north of 30°N have the inter-annual
correlation above 0.5 ($p < 0.1$), 86% of which are located between 45-90°N. The
better correlated $D_{SG}^G$ and $D_{SG}^M$ time series to the north of 45°N than in lower latitudes
implies agreed inter-annual variation of $D_{SG}^G$ and $D_{SG}^M$ in this region. In the regions
with well-correlated inter-annual variation, $D_{SG}$ differences between MODIS and
GIMMS still show significant latitudinal characteristics (Figure S1b). In the northern
mid-latitudes, we inferred a later $\overline{D}_{SG}$ using MODIS($9 \pm 6$ days) in 67% of the pixels,
and an earlier $\overline{D}_{SG}$ ($5 \pm 4$ days) in the remaining pixels, as compared to GIMMS. We
also inferred a later $\overline{D}_{SG}$ using MODIS in southern Asia and the eastern United States
as compared to $\overline{D}_{SG}$ using GIMMS. The $D_{SG}^G$ and $D_{SG}^M$ inter-annual variation are
weakly correlated in the southern mid-latitudes, especially in the Eurasia. For those
pixels in the south of mid-latitude, where inter-annual variation of $D_{SG}^G$ and $D_{SG}^M$ are
well correlated, $D_{SG}^M$ advanced $D_{SG}^G$ by 6±5 days (Figure S1b).

Both MODIS and GIMMS agreed in showing that $D_{SG}$ advanced in Northern

Canada, Eastern United States, and Russia, and that $D_{SG}$ delayed in western North
America, parts of Baltic Europe and East Asia (Figure 1c and 1d). In the regions
where $D_{SG}$ advanced, $D_{SG}^G$ advancement was much weaker than $D_{SG}^M$. In the regions
where $D_{SG}$ delayed, the $D_{SG}^G$ delay is much stronger than $D_{SG}^M$. Together, these
differences lead to a delayed continental-scale $D_{SG}^G$ trend in both North America (0.80
days yr$^{-1}$) and Eurasia (0.22 days yr$^{-1}$) at 90% confidence level. MODIS implied no
significant SG shift trend in North American but advanced SG trend of 0.78 days yr$^{-1}$
in Eurasia at 90% confidence level. The differences in $D_{SG}^G$ and $D_{SG}^M$ trend are mainly



in the northwest of North America and east-to-central Eurasia north of 50°N. The
inter-annual variability of $D_{SG}$ anomalies in relative to $\overline{D}_{SG}$ over 2001-2013 indicated
consistent anomaly signs of $D_{SG}$ between MODIS and GIMMS over 30-50°N in North
America. The most remarkable difference in $D_{SG}$ anomaly between MODIS and
GIMMS is in Northern North America (>50°N) where negative $D_{SG}^{G}$ anomalies over
2001-2008 and positive $D_{SG}^{G}$ anomalies thereafter in North America, in opposite to
$D_{SG}^{M}$ anomalies (Figure 2d). In Eurasia, both MODIS and GIMMS indicated anomalies
of advanced $D_{SG}$ in the north of 50°N after 2006 (Figure 2f). A large transition in the
$D_{SG}^{G}$ anomaly occurred around 2000. The transition is particularly remarkable in North
America, which is due to a 5-6 days later mean $D_{SG}$ ($\overline{D}_{SG}^{G}$) over 2001-2013 than that
over 1982-2000 in North America.
**3.2 Preseason climate regulating SG**

The preseason length of temperature control for GIMMS ( $L_{PT}^{G}$ ) and MODIS

( $L_{PT}^{M}$ ) that we inferred from the correlation between $T_m$ and  $D_{SG}$  differed due to the
differences between $D_{SG}^{G}$ and $D_{SG}^{M}$ (Figure 3a, 3b). The spatial pattern of $L_{PT}^{G}$ shows
significant heterogeneity, with $L_{PT}^{G}$ over two months in the regions from Russia to
central Asia in Eurasia and from Alaska to northwestern Canada in North America.
$L_{PT}^{G}$ is 62±38 days for all the valid pixels, while $L_{PT}^{M}$ is usually less than two months,
with the $L_{PT}^{M}$ of 41±31days. Moreover, $L_{PT}^{M}$ is better correlated to $T_m$ during its
corresponding preseason ($P_T^M$)  with North Hemisphere correlation of 0.6±0.2 in
comparison to the correlation between $D_{SG}^{G}$ and $T_m$ during its preseason ($P_T^G$) of
0.3±0.2 (Figure S2a, 2b).

The fraction of the northern mid- to high-latitude land surface with preseason

precipitation control is less than that for temperature control for both GIMMS and

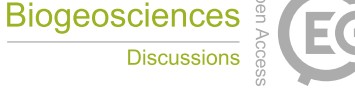


MODIS (Figure 3 and Figure S2). The preseason length of precipitation control for
MODIS ( $L_{PP}^M$ = 56±35 days) is longer than that of temperature control. In contrast,
GIMMS showed relatively shorter preseason length of precipitation control ($L_{PP}^G$ =
45±32 days) than that of temperature control. Although GIMMS showed a larger
fraction of land surface where precipitation correlated to $D_{SG}$ than MODIS, MODIS
and GIMMS showed consistent spatial pattern in both preseason length and
correlations between $P_c$ and $D_{SG}$ (Figure S2c and S2d). The mean PCC is -0.4±0.2 for
both MODIS and GIMMS.

The spatial pattern of the temperature trend in $P_T^M$ and $P_T^G$ over 2001-2013 is

consistent ($r = 0.61$, $p < 0.01$) although the derived preseason length for temperature
control differed for GIMMS and MODIS derived $D_{SG}$ (Figure S3a and S3b). The
majority of both North America and North Eurasia experienced warming of the SG
preseason, while Alaska, the eastern edge of Hudson Bay and the mid-latitudes of
Eurasia (40-60°N) experienced a preseason cooling. The preseason warming trend is
most significant in central Russia and eastern Canada and the cooling trend is most
significant in part of Central Asia and central to eastern China. The maximum
preseason warming trend is about 0.6 °C yr$^{-1}$ in central Russia. The precipitation trend
in the preseason is insignificant and more heterogeneous as compared to the
temperature trend for both $P_P^M$ and $P_P^G$ (Figure S3c and S3d).  The spatial pattern of
the precipitation trend in $P_P^M$ and $P_P^G$ are also less correlated  ($r = 0.40$, $p < 0.01$) than
that of temperature trend. Wetting of the preseason occurred in mid to east of the
United States, Western Canada, Northern Norway and Northwestern Russia. The
largest value of the wetting trend is about 7 mm yr$^{-1}$. Drying preseason only occurred
remarkably in the southeastern the United States and scattered in Eurasia. The pixels
where the largest values of a preseason drying trend is about 4 mm yr$^{-1}$.





**3.3 SG sensitivity to preseason climate**
The fraction of areas in which $D_{SG}^M$ sensitive to $T_m$ and $P_c$ are much larger than
$D_{SG}^G$ (Table S1) and $D_{SG}^M$ are more sensitive to $T_m$ and $P_c$ in relative to $D_{SG}^G$ (Figure 4).
About 43% of the land fraction shows significant sensitivity of $D_{SG}^M$ to $T_m$ ($p < 0.1$)
compared with 13% of the land fraction with significant sensitivity of $D_{SG}^G$ to $T_m$.
About 11% of the land fraction shows significant sensitivity of $D_{SG}^M$ to $P_c$ ($p < 0.1$) as
compared with 3% of the land fraction with significant sensitivity of $D_{SG}^G$ to $P_c$. The
sensitivity of $D_{SG}^M$ to $T_m$ is most significant in the mid- to high-latitudes (Figure 4b)
whereas the sensitivity of $D_{SG}^M$ to $P_c$ is scattered (Figure 4d). The sensitivity  of
MODIS SG to precipitation is 0.23±0.18 days advancement per percent of
precipitation increase. Due to the weak SG-precipitation coupling and sensitivity, we
only analyzed biome-scale $D_{SG}$ to $T_m$ sensitivity (Figure 5). The difference between
the sensitivity of $D_{SG}$ to $T_m$ as inferred by MODIS versus GIMMS is less in forest
biomes, even though $D_{SG}^M$ is more sensitive to $T_m$ in all the biomes in relative to $D_{SG}^G$.
The differences in $D_{SG}$ to $T_m$ sensitivity are especially significant in northern biomes.
For example, sensitivity of $D_{SG}^M$ to $T_m$ in open shrublands, northern grasslands, and
permanent wetlands are 50% higher than sensitivity of $D_{SG}^G$ to $T_m$  in these biomes.
As the GIMMS NDVI product extends as far back as the early 1980s, we also
performed the comparison of $D_{SG}^G$ to $T_m$ sensitivity over two periods. $D_{SG}^G$ to $T_m$
sensitivity was analyzed with the same method in section 2, but between the period
spanning 1988 and 2000.  This has the same length of time (13 years) as the later
analysis period of 2001-2013. The fraction of area where $D_{SG}^G$ shift in response to $T_m$
and $P_c$ is reduced in the period 2001-2013 as compared to the earlier 1988-2000
(Table S1). Most of the biomes show a slightly increased sensitivity of $D_{SG}^G$ to $T_m$ in
the later period, as compared to that over 1988-2000, with the highest increase in the



northern grasslands (44.6%) and open shrublands (41.2%) (Figure 5a). The sensitivity
of $D_{SG}^G$ to $T_m$ is relatively stable in southern grasslands. Exceptionally, the sensitivity
of $D_{SG}^G$ to $T_m$ declined by 1.4 days °C$^{-1}$ for deciduous broadleaf forests and 0.1
days °C$^{-1}$ for mixed forests; this represents a reduced sensitivity of 33.7% and 3.4%
respectively. The inter-biome variation of the sensitivity of $D_{SG}^G$ to $T_m$ is stable ($r =$
0.90, $p < 0.001$) over the two periods (Figure 5b).

## 4. Discussion

### 4.1 SG mean state and trend

We analyzed MODIS and GIMMS NDVI products to infer spring greenup dates
and their responses to preseason climate over the period 2000-2013. Inter-annual
variation of greenup date as inferred from MODIS and GIMMS are well correlated
north of 45ºN (86% of the pixels with $r > 0.5$ and $p < 0.1$). But in these regions, we
tend to infer a later greenup time using MODIS than GIMMS NDVI. This may be
contributed by the evergreen vegetation and the influences of snow cover on the
pixels. The snow cover led to NDVI gaps during the dormancy season. As a result, the
time series of NDVI cannot be adequately fitted during the transitional snow melting
and vegetation greening season (Zhou et al., 2015). We filled the snow-flagged
MODIS NDVI with NDVI from previous period without snow contamination,
whereas GIMMS NDVI was filled with average seasonal profile or spline
interpolation (Pinzon and Tucker, 2014).  Our MODIS filling potentially
underestimate the NDVI during the transition season. In high latitudes with
seasonal snowpack, the beginning of the growing season is often determined by
snowmelt rather than temperature (Semenchuk et al., 2016).  The study over
Yamal Peninsula revealed that spring greenup date is almost the same as snow-end
date between 70.0-73.5ºN (Zeng and Jia, 2013), so that the snow cover affects the



identification of vegetation greenup. In the northern high latitudes at the selected
locations in Canada and Sweden, even if the pixels influenced from snow cover are
excluded, MODIS NDVI is lower than GIMMS NDVI in the dormant season
(Fensholt and Proud, 2012). This can make an explanation to the late transition from
dormant season to growing season by MODIS.

We inferred a heterogeneous trend in SG using both MODIS and GIMMS, but the

sign and magnitude of the SG shift varies between MODIS and GIMMS. The main
difference between the trend in SG as inferred by MODIS and GIMMS is in Alaska
and Siberia, which lead to the main uncertainties in the NDVI derived SG trend in the
northern high latitudes.  The significant GIMMS SG delay in Alaska and mid-latitude
Eurasia lead in general to a delay in SG in North America and Eurasia. In contrast, we
inferred a delay in SG using MODIS in southern Alaska and eastern Canada offset SG
advancement in eastern the United States and Canada, resulting in insignificant SG
trend in North America. Significant SG advancement in Siberia resulted in strong SG
advance in Eurasia. Even so, MODIS and GIMMS showed large inter-annual
variability of SG anomalies in relative to the mean SG over 2001-2013 and the signs
of the anomalies are consistent in between 30ºN and 50ºN. MODIS NDVI inferred
mean SG advancement of 0.96 days year$^{-1}$ between 52-75ºN over 2001-2013 at 90%
confidence level in our results overwhelmed the MODIS snow-end date advancement
of 0.37 days year$^{-1}$ in this region over 2001-2014 (Chen et al., 2015). The lagged
snow phenology advancement implies that snow complication in determine SG in the
cold regions is still present at a warmer climate.
**4.2 SG dates sensitivities to climate**

The SG to preseason climate sensitivity by MODIS and GIMMS showed

varied degree of vegetation-climate seasonal coupling. The higher correlation between



MODIS SG and preseason temperature indicates stronger MODIS SG-climate
relationships. The stronger MODIS NDVI to temperature correlation than GIMMS
NDVI was reported in central Europe, where the correlation between temperature and
August NDVI anomalies were analyzed (Kern et al., 2016). The stronger SG-
temperature coupling than precipitation is consistent with our previous study of SG to
climate sensitivity over 1982-2005 (Xu et al., 2018). MODIS inferred stronger SG-
temperature sensitivity in the northern boreal and Arctic biomes can be explained by
the site-level observation that temperature sensitivity of phenology is greater in colder,
higher latitude sites than in warmer regions (Prevéy et al., 2017). At the colder sites,
the small changes in temperature may constitute greater relative changes in thermal
budget (Oberbauer et al*.,* 2013), so that the warming impacts on vegetation are
amplified. This explanation is not applicable to the GIMMS NDVI inferred SG
response to temperature that vegetation with earlier growing season is more sensitive
to temperature (Shen et al., 2014).

The sensitivity of GIMMS SG to temperature increased over 2001-2013 in

relative to that over 1988-2000. Our results showed SG to temperature sensitivity
increased most significantly in Arctic grassland (44.6%), followed by other boreal
biomes (open shrubland (41.2%), permanent wetland (35.9%), woody savanna (31.1%)
and deciduous needleleaf forest (17.6%)). The magnitudes of enhanced sensitivity are
even larger when we compare 2001-2013 SG-temperature sensitivity with a longer
period over 1982-2005 (Xu et al., 2018). Compare with the period 1982-2005, SG-
temperature sensitivity of the northern biomes (deciduous needleleaf forest, woody
savanna, open shrublands and permanent wetlands) all increased more than 50% over
2001-2013 with stable inter-biome sensitivity variation ($r = 0.91$, $p < 0.01$).



The increased sensitivity of SG to temperature for boreal biomes has not been
well investigated. In the contrary, temperature sensitivity of spring greenup may
decline under warmer climate because (1) insufficient winter chilling may delay the
spring greenup in spite of continued spring warming (Yu et al., 2010), (2) when
spring greenup starts earlier, shorter photoperiod can limit the potential of leaf
development (Chmielewski & Götz, 2016), (3) greenup may respond nonlinearly to
temperature and be saturated at a high temperature (Caffarra & Donnelly, 2011), and
(4) under warmer condition, the preseason duration of thermal forcing can be reduced,
which declines the SG-temperature sensitivity (Güsewell et al., 2017). The vegetation
growth (represented by NDVI) to temperature sensitivity was reported declining in
the growing season (April-October) based on GIMMS NDVI over 1982-2012 linked
to water stress (Piao et al., 2014). In temperate ecosystems, the lower NDVI to
temperature sensitivity coincidently occurred with increased drought events.  While in
the arctic ecosystem, the lowered sensitivity of NDVI to temperature may be
explained by increases in heat waves because the physiological response of
photosynthesis to temperature is nonlinear with lower sensitivity under warmer
conditions (Piao et al., 2014). The higher interannual temperature variability can also
cause higher variations in water supply, thus the declined coupling between
vegetation growth and interannual variability of growing season temperature,
generally in semiarid regions (Wu et al., 2017). The wetting preseason in mid to east
of the United States, Western Canada, Northern land along Norway and Northwestern
Russia may partly enhanced SG-temperature if the enhancement is validated.
**4.3 Uncertainties in SG as derived by MODIS and GIMMS NDVI**
With SG as inferred using GIMMS over the period 1988-2000 and as inferred
using MODIS over 2001-2013, we found that the trend is advanced continuously in



response to a continuing trend in preseason warming. The uncertainties in the SG
trend and its climatic sensitivity arise when SG as inferred using MODIS and GIMMS
are compared together over the period 2001-2013. Wang et al.(2016) and Zhang et al.
(2013) proposed that quality issues may present in GIMMS NDVI, which can bias
vegetation growth sensitivity and growth trend. Instead of using continuous GIMMS
SG over 1982-2011, Zhang et al. (2013) merged datasets of GIMMS SG over 1982-
2000 and SPOT-VGT SG over 2001-2011 to detect SG trend due to data quality
issues with GIMMS NDVI in most parts of western Tibetan Plateau, according to the
findings of opposite GIMMS SG trend to SPOT-VGT and MODIS SG trend over the
period 2001-2006. With this merged data record, the SG trend continuously advanced
in Tibetan Plateau over 1982-2011. This result is consistent with the SG trend derived
from tree-ring data (Yang et al., 2017). On the contrary, continuous GIMMS SG over
1982-2006 inferred delayed SG trend after mid-1990s over Tibetan Plateau (Yu et al.,
2010). At the North Hemisphere scale, GIMMS SG (1982-2008) showed significant
decadal variation and declining SG shift: advanced 5.2 days over 1982-1999, but only
advanced 0.2 days over 2000-2008 (Jeong et al., 2011). However, the merged
GIMMS (1982-2006) and MODIS (2002-2012) showed SG shift over 2002-2012 (-6
days decade$^{-1}$) is about three times larger than that over 1982-2002 (-2 days decade$^{-1}$),
which is interpreted as enhanced SG advancement and its response to temperature
over time (Wang et al., 2016). For the varied timing of SG derived from different
products, Zhang et al. (2017) suggested intersensor calibrations to reduce the
difference between vegetation index products and exclusion of the low quality
phonology timing.

These SG shift uncertainties after 2000 are more likely to be explained by the

differences in the NDVI products that implied the opposite SG trend, anomalies north



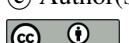

of 50ºN and biome-scale SG-temperature sensitivities. The spectrum range difference
of MODIS and AVHRR sensor channels is a main contribute to the NDVI differences.
MODIS NDVI is derived from bands 1(620-670nm) and 2 (841-876nm) of the
MODIS on board NASA's Terra satellite whereas GIMMS NDVI is derived from
bands 1(580-680nm) and 2 (725-1100nm) of AVHRR. The large GIMMS SG
anomaly transition around 2000 may be associated with the sensor transition from
AVHRR/2 to AVHRR/3, although among-instrument AVHRR calibration were
conducted with NDVI data derived from Sea-Viewing Wide Field-of-view Sensor
(SeaWiFS) (Pinzon et al., 2014). The calibration with SeaWiFS is considered as an
improvement of GIMMS NDVI in the very northern latitudes (Marshall et al, 2016).
Even so, the data issues associated with sensor transition, such as (1) satellite signal
degradation through lifetime, (2) band design, (3) effect of maximum value composite
(MVC) and (4) replacement of satellites in NOAA series, potentially influence the
interpretation of the SG trend and its sensitivity to climate drivers.
**5. Conclusions**

We compare the MODIS and GIMMS NDVI inferred time of spring greenup

and its response to preseason climate over 2001-2013. We infer a spring greenup
delay using GIMMS NDVI in both North America (0.80 days yr$^{-1}$) and Eurasia (0.22
days yr$^{-1}$), whereas, using MODIS NDVI, we infer no significant spring greenup shift
in North American and an advanced SG trend of 0.78 days yr$^{-1}$ in Eurasia. The
differences in MODIS and GIMMS inferred spring greenup trend are mainly in
northern high latitude (>50ºN). The differences are implied by opposite anomalies in
the time of spring greenup in North America and a large GIMMS inferred spring
greenup transition around 2000 that maybe explained by data issues associated with
the sensor transition from AVHRR/2 to AVHRR/3, including (1) satellite signal





degradation through lifetime, (2) band design, (3) effect of maximum value
composite (MVC) and (4) replacement of satellites in NOAA series. Temperature is
the primary climate driver of the time of spring greenup for both MODIS and GIMMS,
although MODIS inferred both a stronger sensitivity and correlation between SG and
temperature. The opposing trends of SG as inferred using MODIS and GIMMS
resulted in differing SG to temperature sensitivity across biomes (-3.6±0.7 days °C$^{-1}$
for MODIS and 2.2 ± 0.8 days °C$^{-1}$ for GIMMS). Using GIMMS, we inferred that the
sensitivity of greenup to temperature, which increases over time for Arctic and boreal
biomes, cannot be well explained by the mechanisms regulating the sensitivity of SG
under a warming climate. This result requires further investigation. Our results
suggest the importance of snow-vegetation interactions in high latitude vegetation
monitoring and inter-validation of multiple datasets to better assess vegetation
dynamics.



**Acknowledgements**
This study is funded by the Strategic Priority Research Program of the Chinese
Academy of Sciences, CASEarth (XDA19070203), grants from CAS Pioneer
Hundred Talents Program and Natural Science Foundation of China (NSFC
#41590853). We acknowledge support from the U.S. Department of Energy, Office of
Science, Biological and Environmental Research, Regional and Global Climate
Modeling Program through the RUBISCO Scientific Focus Area under contract
DE-AC02-05CH11231 to Lawrence Berkeley National Laboratory.



**The authors declare no conflicts of interest.**

**Supplements**
**Figure S1**
**Figure S2**
**Figure S3**
**Table S1**



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





**Figure Captions:**
Figure 1. The GIMMS (a) and MODIS (b) inferred mean Julian date of spring
greenup ($\overline{D}_{SG}$, day of year) over 2001-2013 and GIMMS (c) and MODIS (d) inferred
trend of spring greenup date ($D_{SG}$) over 2001-2013(days yr$^{-1}$).
Figure 2 Anomalies of spring greenup date for mid-latitude (30-50ºN, a, c, e) and high
latitude (>50ºN, b, d, f) in relative to mean $D_{SG}$ over 2001-2013 for GIMMS and
MODIS.
Figure 3 Mean preseason length of temperature control corresponding to GIMMS
spring greenup ($\overline{L}_{PT}^{G}$, days) and MODIS spring greenup ($\overline{L}_{PT}^{M}$, days) and mean
preseason length of precipitation control corresponding to GIMMS spring greenup
($\overline{L}_{PP}^{G}$, days) and MODIS greenup ($\overline{L}_{PP}^{M}$, days).
Figure 4 Spring greenup sensitivity to preseason temperature (days °C-1) for GIMMS
(a) and MODIS (b) and spring greenup sensitivity to preseason precipitation (days %$^{-1}$
of precipitation increases) for GIMMS (c) and MODIS (d).
Figure 5 The comparison of inter-biome SG sensitivity to preseason temperature for
IGBP land cover types for GIMMS over 1982-2005 and 2001-2013 and MODIS over
2001-2013. We used the IGBP land cover classification for 9 biomes in 2012:
Evergreen Needleleaf Forest (ENF), Deciduous Needleleaf Forest (DNF), Deciduous
Broadleaf forest (DBF), Mixed Forest (MF), Open Shrublands (OS), Woody
Savannas (WS), Grassland (GL), Permanent Wetland (PW), and Cropland (CP). We
distinguish the Arctic grassland to the north of 60°N (GLN), from temperate grassland
in the south (GLS) due to their expected differences in climate and controls on
phenology.





Table 1  The spring greenup shift (days per decade) as inferred from Normalized Difference Vegetation Index (NDVI) from satellite data

| NDVI Data | Period | Region | Shift (days decade$^{-1}$) | Reference |
|---|---|---|---|---|
| PAL | 1981-1991 | >=40N | -8 | Myneni et al., 1997 |
| GIMMS | 1981-1999 | Eurasia | -3.3 | Zhou et al., 2001 |
| GIMMS | 1981-1999 | N. America | -4.4 | Zhou et al., 2001 |
| AVHRR | 1982-1991 | 45-75 | -6.2 | Tucker et al., 2001 |
| AVHRR | 1992-1999 | 45-75 | -2.4 | Tucker et al., 2001 |
| AVHRR | 1982-1990 | Inner Mongolia | 0 | Lee et al., 2002 |
| PAL | 1982-2001 | Europe | -5.4 | Stockli and Vidale, 2004 |
| PAL | 1985-1999 | N. America | -6.6 | de Beurs and Henebry, 2005 |
| PAL | 1985-2000 | Eurasia | -4.5 | de Beurs and Henebry, 2005 |
| GIMMS | 1982-1999 | Temperate China | -7.9 | Piao et al., 2006 |
| PAL | 1982-1999 | East Asia | -7 | Jeong et al., 2009 |
| GIMMS | 1982-2003 | Global | -3.8 | Julien & Sobrino, 2009 |
| GIMMS | 1982-2006 | Fennoscandia | -2.7 | Karlsen et al., 2009 |
| GIMMS | 1982-1999 | N. Hemisphere | -2.9 | Jeong et al., 2011 |
| GIMMS | 2002-2008 | N. Hemisphere | -0.3 | Jeong et al., 2011 |
| MODIS | 2000-2010 | >60N, Arctic | -4.7 | Zeng et al., 2011 |
| MODIS | 2000-2010 | >60N, N. America | -11.5 | Zeng et al., 2011 |
| MODIS | 2000-2010 | >60N, Eurasia | -2.7 | Zeng et al., 2011 |
| GIMMS | 1982-2008 | >60N, Arctic | -0.5 | Zeng et al., 2011 |
| GIMMS | 1982-2008 | >60N, N. America | -0.8 | Zeng et al., 2011 |
| GIMMS | 1982-2008 | >60N, Eurasia | -0.3 | Zeng et al., 2011 |
| GIMMS SPOT-VGT | 1982-2011 | Tibetan Plateau | -10.4 | Zhang et al., 2013 |
| GIMMS | 1982-2011 | Fennoscandia | -11.8 | Høgda et al., 2013 |
| MODIS | 2001-2012 | U.S. | -4.8 | Keenan et al., 2014 |
| MODIS | 2002-2014 | Inner Mongolia | -4.5 | Gong et al., 2015 |
| GIMMS | 1982-2011 | U.S. Great Basin | -0.1 | Tang et al., 2015 |
| GIMMS | 1982-2002 | N. Hemisphere | -1.9 | Wang et al., 2016 |
| MODIS | 2002-2012 | N. Hemisphere | -5.9 | Wang et al., 2016 |
| GIMMS | 1982-2012 | Tibetan Plateau | 0 | Ding et al., 2016 |

MODIS: Moderate Resolution Imaging Spectroradiometer
AVHRR: Advanced Very High Resolution Radiometer
GIMMS: Global Inventory Modeling and Mapping Studies
PAL: Pathfinder AVHRR Land
GAC: Global area cover





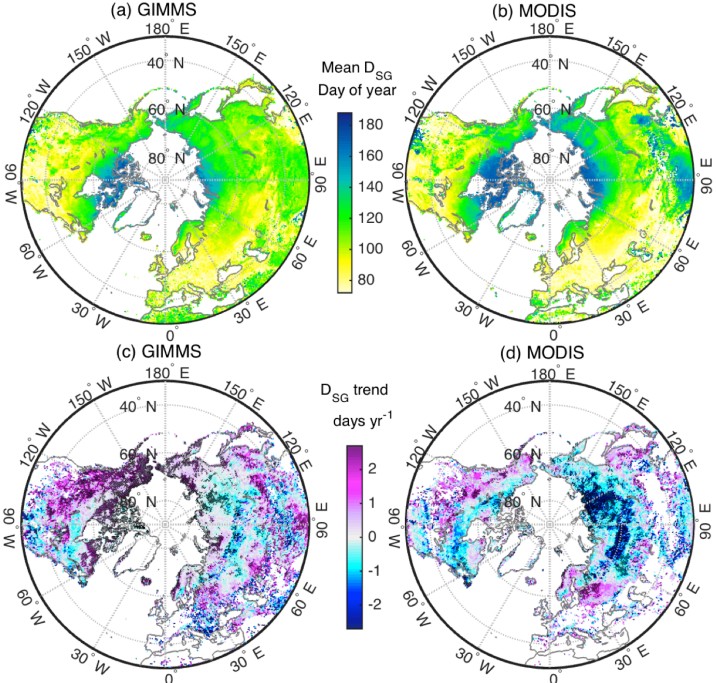

Figure 1. The GIMMS (a) and MODIS (b) inferred mean Julian date of spring greenup ($\overline{D}_{SG}$, day of year) over 2001-2013 and GIMMS (c) and MODIS (d) inferred trend of spring greenup date ($D_{SG}$) over 2001-2013(days yr$^{-1}$).





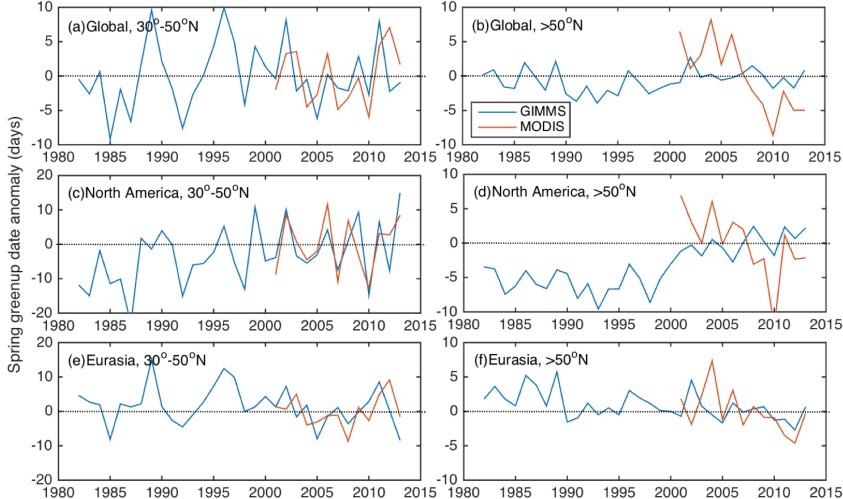

Figure 2 Anomalies of spring greenup date for mid-latitude (30-50° N, a, c, e)
and high latitude (>50° N, b, d, f) in relative to mean $D_{SG}$ over 2001-2013 for
GIMMS and MODIS.





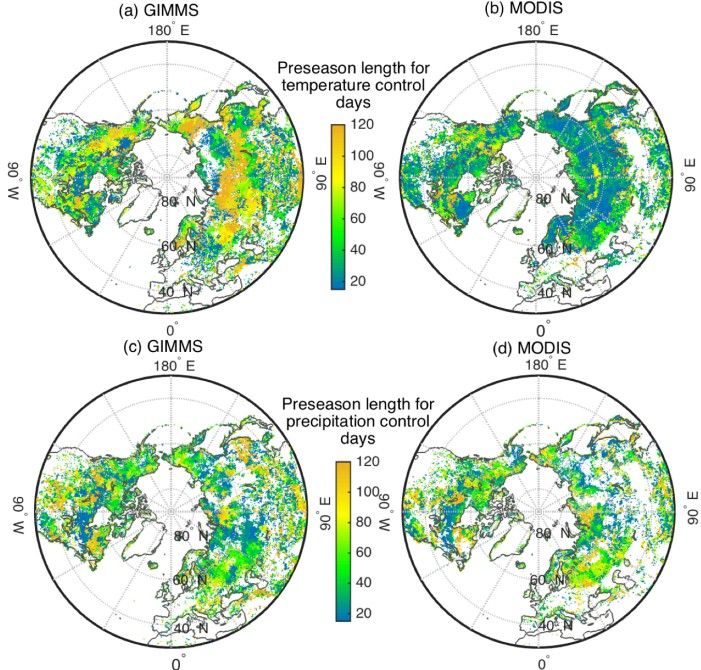

Figure 3 Mean preseason length of temperature control corresponding to GIMMS spring greenup ($\overline{L}_{PT}^{G}$, days) and MODIS spring greenup ($\overline{L}_{PT}^{M}$, days) and mean preseason length of precipitation control corresponding to GIMMS spring greenup ($\overline{L}_{PP}^{G}$, days) and MODIS greenup ($\overline{L}_{PP}^{M}$, days).



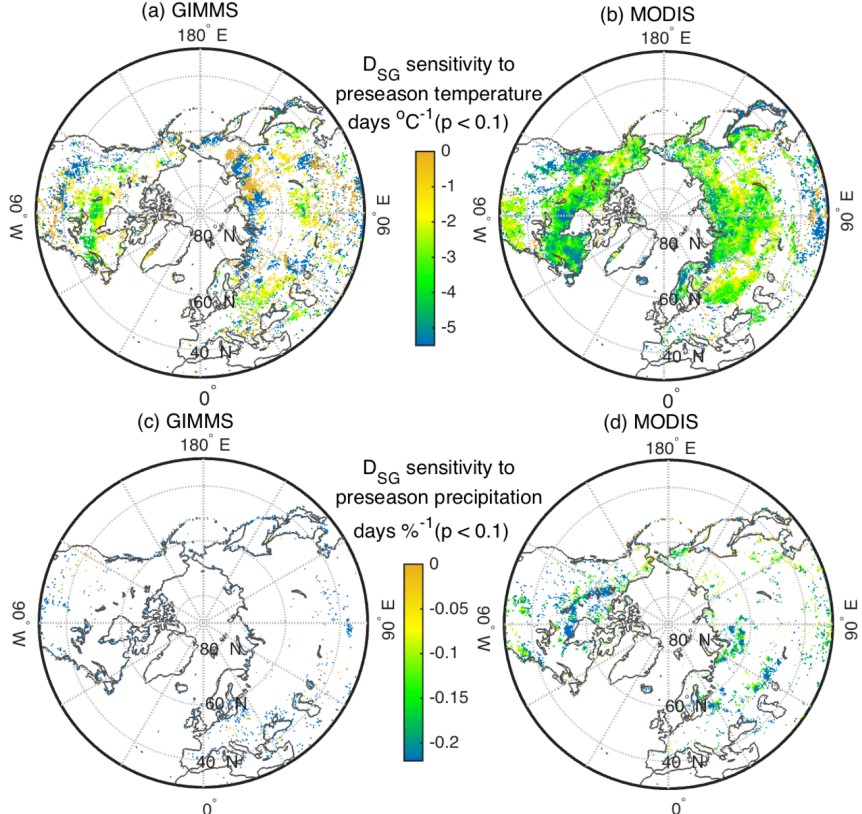

Figure 4 Spring greenup sensitivity to preseason temperature (days °C-1) for
GIMMS (a) and MODIS (b) and spring greenup sensitivity to preseason
precipitation (days %⁻¹ of precipitation increases) for GIMMS (c) and MODIS
(d).





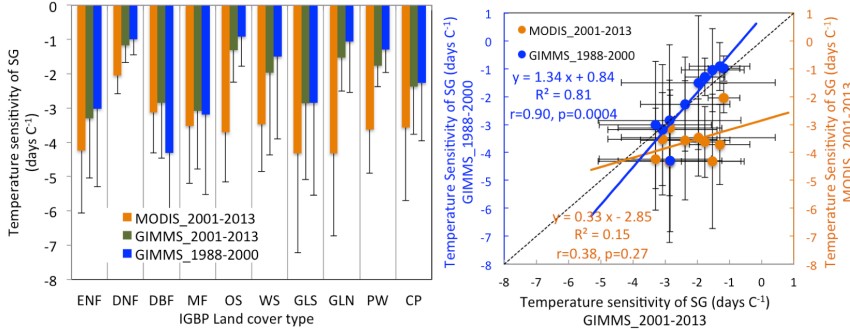

Figure 5 The comparison of inter-biome SG sensitivity to preseason
temperature for IGBP land cover types for GIMMS over 1982-2005 and 2001-
2013 and MODIS over 2001-2013. We used the IGBP land cover classification
for 9 biomes in 2012: Evergreen Needleleaf Forest (ENF), Deciduous
Needleleaf Forest (DNF), Deciduous Broadleaf forest (DBF), Mixed Forest
(MF), Open Shrublands (OS), Woody Savannas (WS), Grassland (GL),
Permanent Wetland (PW), and Cropland (CP). We distinguish the Arctic
grassland to the north of 60°N (GLN), from temperate grassland in the south
(GLS) due to their expected differences in climate and controls on phenology.