# Peer review of "Spring phenology and phenology-climate links inferred from two remotely sensed vegetation indices across regions and biomes"

_Biogeosciences, 2018_

## Referee Comment (RC1) · Anonymous Referee #1 · 14 Aug 2018

Xu and coauthors used two different NDVI phenology metrics to determent the dynamics of spring green-up dates, and studied the correlation between greenup and preseason T and precipitations. Large scale phenology study has been a hot pot in the global change ecology study, and comparison studies in the RS-based phenology dates have been investigated several times recent yearsïijŇbut did not find consistent results that single method could be perfectly used to extract phenology data from the RS data series and therefore multiple methods that were used to extract phenology transition dates were recommend. This study focused on the RS based phenology and try to understand the difference in the RS-based phenology dates between two NDVI metrics, it would be a good contribution to understand the RS phenology, but I have

several major comments that hopefully can help to improve this analysis. 1) In this study, the authors focused on the AVHRR and MODIS, and found that, over the period 2001-2013, difference in magnitude and sign in spring phenology dates between these two dataset. Even, over the long period, i.e. 1983-2014 using AVHRR, globally delayed spring phenology dates were reported, which is different from the in situ data, as well as many regional phenology studies.Considering large variation in spring phenology, a 10yrs trends may holds large uncertainty in trend analysis. Furthermore, only one single methods, i.e. piecewise logistic method, might be also generate large uncertainty in the green-up extraction. Therefore, I would suggest to apply multiple methods to extract green-up dates. Since large uncertianty in the gridded climate data, validated study using another climate dataset would be suggested, and the reuslts could be put in the appendix. 2) Wrong estimation in preseason and T/Precipitation sensitivity. About the preseason issues, from the figure 3, very large difference preseaon-T were found between MODIS and AVHRR, but using same climate dataset and very similar green-up data, except Tibet and Polar regions, is it possible the large difference related to the statistics method? i.e. the DF is too small, i.e. 2001-2014, and could not be perfectly used to determine the preseason length. Or other climatic issues might affect the phenology process, and thus the effect of precipitation and radiation should be excluded from the preseason estimation. Anyway, the large difference in the preseason estimation is wired, and it would be substantially affect the estimation in the sensitivity estimation. In the temperature/precipitation sensitivity of phenology, only significant relationships were recorded and mapped, but the percentage should also be provided. From the results and figure4, it seems quiet large percentage pixels were removed. if 90% were insignificant and removed, then a mean values across the 10% in T-sensitivity would be nonsense. I would suggest to provide all data, both insignificant and sig correlations, and calculate the mean values and provide the percentage of sigs. 3) the reference should be provided in many arguments, such as L37, what's means of several changes? need references; L114, need references, also why moisture? generally by T and photoperiod, rather air or soil moisture..

---

## Referee Comment (RC2) · Anonymous Referee #2 · 20 Aug 2018

Summary:

Xu et al. manuscript presents a comparison of spring greenup (SG) and its sensitivity to temperature and precipitation over the northern hemisphere inferred from two remote sensing NDVI products (MODIS and GIMMS) over the period 2001-2013. They aim at exploring the uncertainties in NDVI SG trends and sensitivity to climate. They conclude that both products are consistent in mid latitudes both for SG and its sensitivity to climate, but show different magnitude and trends in high latitudes.

General Comments:

The analysis performed in this study is timely, but the main message and the novelty

of the study remain unclear.

First, the authors did not really assessed the uncertainties induced by NDVI products and the approach used. Previous studies already highlighted differences in SG and its temporal trend estimated by several approaches and different NDVI products (Chang et al. 2016, Wang at am. 2015, Ding et al. 2015 for example). It is already known that main uncertainties in estimating SG are found in high latitudes. Xu et al. went one step further by comparing SG sensitivity to climate between products but mainly concluded about observed differences, not the uncertainty behind, which in the end led to the same conclusion than previous studies. Because differences in SG estimates leads to differences in pre-season length, the authors compared sensitivities that are not really comparable. The uncertainty in sensitivity to climate results from the propagation of the uncertainty in SG estimates, however these aspects are poorly discussed in the manuscript.

Secondly, the methodology suffers from major flaws. Only one method is used to smooth and fit the data while previous studies highlighted a strong impact of the smoothing method (Atkinson et al. 2012) and approach used to estimate SG. More-over we don't have information about the performance of the approach. The authors did not take into account partial correlations between temperature and precipitation, thus leading to weak interpretations of the results (Fu et al. 2014). Finally the filtering of data performed in the analysis is sometimes unjustified or incomplete (see specific comments).

The authors should refine their research question to be in adequation with their approach or go deeper in the analysis of uncertainty propagation from NDVI products to the estimation of climate sensitivity of SG.

Specific Comments:

L.96: the paper from cong et al. is not an evidence that methods in estimating SG has no impact on resulting trends and sensitivity. Especially for sensitivity to climate that

requires the estimation of the pre-season length, there will be a propagation of errors that will influence final results.

L.135: now CRU-NCEP v.8 is extended to 2017. Remove "recently extended"

L.145: maybe use the median value.

L.142 & 179: Is 15days observation fine enough to estimate sensitivity to climate change properly? Moreover, is it significant to estimate the pre-season length with a 3days step when observations are performed every 15 days?

L.179-183:explain why?

L.184: the authors should use partial correlations to take into account co-variations of climate.

L.187 & 195: why removing positive correlations? Several studies highlighted different behaviours according to species and regions (Zhang et al. 2016 for example). By removing positive correlations you start the analysis by assuming that vegetation respond all the time negatively to climate change which is not true. If the aim is to compare both products the authors should keep all the information available.

L.194: OLS or SMA regression. In this case SMA regression are more appropriate

L.195: why not excluding non vegetated pixels? It would improve the analysis.

L.195: to avoid a bias due to the number of significant pixels, the authors should compare only pixels for which significant sensitivities can be estimated for both products.

L243: check partial correlations between SG temperature and precipitation.

L247: a significant correlation does not mean a control. Please reformulate

L.252-256: how does it relate to changes in SG?

L.271: Does +/- 7 or 4mm means a significant change in precipitation? .

L.275: because we don t have the same pre-season length it is difficult to conclude.

L.280: why it is not responsive to precipitation?

L.283: percent compared to which value?

L.300: interesting result. It is consistent with field observations over Europe (Fu et al. 2015)

L.312: recent studies showed that CCI is better than NDVI in detecting phonological changes for evergreen (Gamon et al. 2016). That may explain the behaviour of evergreen vegetation in this study.

Technical comments:

Figures 2abc are not cited in the text

As you compare both products figure s1 is more relevant than figure 1. Try to use absolute or relative comparisons in the main figures and put absolute values in supp, also for fig3. Moreover the scale make it difficult to see where differences are null.

Atkinson, Peter M., et al. "Inter-comparison of four models for smoothing satellite sensor time-series data to estimate vegetation phenology." Remote sensing of environment 123 (2012): 400-417.

Wang C, Cao R, Chen J, Rao Y, Tang Y. 2015. Temperature sensitivity of spring vegetation phenology correlates to within-spring warming speed over the Northern Hemisphere. Ecol Indic. 50:62–68.

Ding M, Li L, Zhang Y, Sun X, Liu L, Gao J, Wang Z, Li Y. 2015. Start of vegetation growing season on the Tibetan Plateau inferred from multiple methods based on GIMMS and SPOT NDVI data. J Geog Sci. 25:131–148. Better consistency with observations is the threshold

Chang, Qing & Zhang, Jiahua & Wenzhe, Jiao & Yao, Fengmei. (2016). A

comparative analysis of the NDVIg and NDVI3g in monitoring vegetation phenology changes in the Northern Hemisphere. Geocarto International. 33. 1-20. 10.1080/10106049.2016.1222633.

Zhang, H.,W. Yuan, S. Liu, andW. Dong. 2015. Divergent responses of leaf phenology to changing temperature among plant species and geographical regions. Ecosphere 6(12):250. http://dx.doi.org/10.1890/ES15-00223.1

Gamon, John A., et al. "A remotely sensed pigment index reveals photosynthetic phenology in evergreen conifers." Proceedings of the National Academy of Sciences 113.46 (2016): 13087-13092.

---

## Referee Comment (RC3) · Anonymous Referee #3 · 31 Aug 2018

Summary

The manuscript compares the spring greenup date (SG) obtained using the method of Zhang et al. (2003) applied to the GIMMS dataset and the MODIS MOD13C1 dataset. It also compares the trend of the chronological regression on the same period, and the sensitivity of the SG to the climate before the SG. The results are completed by the analysis of the trend before the launch of MODIS.

General evaluation

The current manuscript has a nice first objective which is to explore the impact of the input remote sensing dataset to which the SG algorithm is applied. However it requires

more analysis to achieve this objective. If the objective is to evaluate the impact of the choice of the input dataset then the causes of the differences should be the target. This would allows to increase the confidence in the trend analysis based on the two datasets. The suggestion on the role of the transition from AVHRR2 to AVHRR3 should also be explored in depth as this is quite an interesting opening. In the following I suggest some changes and some previous articles that should be considered. The second objective on the sensitivity of SG to climate cannot be achieved without improvements on the work on the SG determination, as there is strong uncertainty on the SG.

Major comments

1/ The results report differences in the SG obtained with MODIS and with AVHRR and in the trend. The manuscript suggests possible explanations but does not attempt to determine the reasons clearly. In-depth exploration of the causes of these differences should be carried out as it is the aim of the manuscript. Possible sources of uncertainties, including spatial resolution changes with incidence angle, preprocessing, processing are explored in :

Helman, D. (2018) Land surface phenology: What do we really "see" from space? Science of The Total Environment, 618, 665 – 673.

Moreover, a key issue is the snowmelt. It is well-known that the detected SG is related to snowmelt and not to vegetation if the snow is not correctly treated (Moulin et al. 1997, Shabanov et al. 2002). Solutions have been proposed (Suzuki et al. 2003, Delbart et al. 2005, 2006, Thomson et al. 2015, Jin et al. 2017 for examples). In the current manuscript the snow issue is treated differently between the two dataset preprocessing, thus the results differ. It is necessary to assess the uncertainty of the two SG datasets through a comparison to ground observations of leaf expansion, in order to analyse the impact of the snow rejection methods.

Delbart, N., Kergoat, L., Le Toan, T., Lhermitte, J., & Picard, G. (2005) Determination of phenological dates in boreal regions using normalized difference water index. Remote

[Figure]

Sensing of Environment, 97, 26–38.

Delbart, N., Le Toan, T., Kergoat, L., & Fedotova, V. (2006) Remote sensing of spring phenology in boreal regions: A free of snow-effect method using NOAA-AVHRR and SPOT-VGT data (1982-2004). Remote Sensing of Environment, 101, 52–62.

Jin, H., Jönsson, A.M., Bolmgren, K., Langvall, O., & Eklundh, L. (2017) Disentangling remotely-sensed plant phenology and snow seasonality at northern Europe using MODIS and the plant phenology index. Remote Sensing of Environment, 198, 203–212.

Moulin, S., Kergoat, L., Viovy, N., & Dedieu, G. (1997) Global-scale assessment of vegetation phenology using NOAA/AVHRR satellite measurements. Journal of Climate, 10, 1154–1170.

Shabanov, N.V., Zhou, L., Knyazikhin, Y., Myneni, R.B., & Tucker, C.J. (2002) Analysis of interannual changes in northern vegetation activity observed in AVHRR data from 1981 to 1994. IEEE Transactions on Geoscience and Remote Sensing, 40, 115–130.

Suzuki, R., Nomaki, T., & Yasunari, T. (2003) West-east contrast of phenology and climate in northern Asia revealed using a remotely sensed vegetation index. International Journal of Biometeorology, 47, 126–138.

Thompson, B.G. (2015) Using phase-spaces to characterize land surface phenology in a seasonally snow-covered landscape. Remote Sensing of Environment, 166, 178–190.

2/ The pre-season length differs when so the variations of the climate variables differ. Thus the trend of the preseason climate from the two SG datasets should not be compared.

3/ The trends are reported if the p-value is less than 0.1. This is not a good value as it is too high. Maximum is generally 0.05, which is very high already.

5/ The GIMMS dataset and the MOD13C1 dataset are composite products with a compositing period of 15 and 16 days. This has a strong impact on SG uncertainty. This is why PAL product should be prefered (10 day composite) of 8-day MODIS dataset. The effect of the compositing period duration must be explored.

6/ Reported trends should be compared to those from :

Gonsamo, A. & Chen, J.M. (2016) Circumpolar vegetation dynamics product for global change study. Remote Sensing of Environment, 182, 13–26.

Park, T., Ganguly, S., Tømmervik, H., Euskirchen, E.S., Høgda, K.-A., Karlsen, S.R., Brovkin, V., Nemani, R.R., & Myneni, R.B. (2016) Changes in growing season duration and productivity of northern vegetation inferred from long-term remote sensing data. Environmental Research Letters, 11, 084001.

Minor comments

1/ Analysing the sensitivity of SG to temperature through linear correlation is not totally convincing. The phenology models are well known to be unlinear (non linear effects are mentionned in the discussion) and parameterized with thresholds. See for example :

Hänninen, H. (1990) Modelling bud dormancy release in trees from cool and temperate regions. Acta Forestalia Fennica, 213, 1-47.

The consequence is that, for example in an arctic ecosystem, a warming from -15°C to -5°C in March will have no impact on SG whereas a warming from 2°C to 3°C in may would have a strong impact. Thus changes in sensitivity of SG to temperature changes are expected.

2/ The style of the writing is often hard to read and the text should be clarified.

---

## Author Comment (AC1) · 15 Oct 2018

*Responses to Anonymous Referee #1*

Xu and coauthors used two different NDVI phenology metrics to determent the dynamics of spring green-up dates, and studied the correlation between green-up and preseason T and precipitations. Large-scale phenology study has been a hot pot in the global change ecology study, and comparison studies in the RS-based phenology dates have been investigated several times recent years, but did not find consistent results that single method could be perfectly used to extract phenology data from the RS data series and therefore multiple methods that were used to extract phenology transition dates were recommended. This study focused on the RS based phenology and try to understand the difference in the RS-based phenology dates between two NDVI metrics, it would be a good contribution to understand the RS phenology, but I have several major comments that hopefully can help to improve this analysis.

*Authors:* We sincerely thank the reviewer for constructive criticisms and valuable comments. We have edited and rewritten parts of this manuscript. Our responses and detailed edits are indicated below in the point-to-point responses.

1) In this study, the authors focused on the AVHRR and MODIS, and found that, over the period 2001-2013, difference in magnitude and sign in spring phenology dates between these two dataset. Even, over the long period, i.e. 1983-2014 using AVHRR, globally delayed spring phenology dates were reported, which is different from the in situ data, as well
as many regional phenology studies. Considering large variation in spring phenology, a 10yrs trends may holds large uncertainty in trend analysis. Furthermore, only one single methods, i.e. piecewise logistic method, might be also generate large uncertainty in the green-up extraction. Therefore, I would suggest applying multiple

methods to extract green-up dates. Since large uncertainty in the gridded climate data, validated study using another climate dataset would be suggested, and the results could be put in the appendix.

*Authors:* This study was inspired by the different spring phenology trend between AVHRR and MODIS, and between AVHRR and in situ observation. In 1980s and 1990s, AVHRR showed a consistent advanced trend with the in-situ observations, even though the advanced magnitude differs. The main difference was found after 2000, which is the period overlapping with MODIS product for comparison. We agree with the reviewer that a trend analysis over a 13 years is a bit short, however, we compared the estimated mean spring phenology and interannual variability between AVHRR and MODIS NDVI derived SGs. Furthermore, our analysis of spring phenology sensitivity to temperature and precipitation is the interannual response to climate.

We add a paragraph in section 2.4 to elucidate why we use a single method to smooth NDVI time series and derive greenup-date other than multi-methods. The common used regression methods to reconstruct NDVI time series and derive SG include Savitzky-Golay fitting method, spline smoothing, asymmetric Gaussian functions, double logistic function, and harmonic analysis of times series. These methods are valid in fitting NDVI gaps and reducing noise (Hird and McDermid, 2009; Cai et al. 2017), however, can make differences in estimating phonological stages (Cong et al., 2013). It is hard to justify which method is better. In order to reduce the mixed uncertainty of reconstruction methods and NDVI products, here we used one regression method to reconstruct the NDVI series. The double logistic method uses least-square fitting to half growing season (Zhang et al., 2003). It requires no smoothing parameter and is more robust than other methods in estimating

the vegetation seasonal dynamics, when there is no local calibration (Cai et al., 2013).

Hird, J. N., G. J. McDermid (2009). Noise reduction of NDVI time series: An empirical comparison of selected techniques. Remote Sensing of Environment, 113, 248-258.

Cai, Z., P. Jönsson, H. Jin, L. Eklundh (2017). Performance of smoothing method for reconstructing NDVItime-series and estimating vegetation phenology from MODIS data. Remote Sensing, 9, 1271, doi:10.3390/rs9121271.

Cong, N., T. Wang, H. Nan, Y. Ma, X. Wang, R. B. Myneni, S. Piao (2013). Changes in satellite-derived spring vegetation green-up date and its linkage to climate in China from 1982 to 2010: a multi-method analysis. Global Change Biology, 19, 881-891, doi:10.1111/gcb.12077.

Zhang, X., Friedl, M.A., Schaaf, C.B., Strahler, A.H., Hodges, J.C.F., Gao, F., Reed, B.C., Huete, A. (2003). Monitoring vegetation phenology using MODIS. Remote Sensing of Environment, 84, 471–475.

2) Wrong estimation in preseason and T/Precipitation sensitivity.

About the preseason issues, from the figure 3, very large difference preseaon-T were found between MODIS and AVHRR, but using same climate dataset and very similar green-up data, except Tibet and Polar regions, is it possible the large difference related to the statistics method? i.e. the DF is too small, i.e. 2001-2014, and could not be perfectly used to determine the preseason length. Or other climatic issues might affect the phenology process, and thus the effect of precipitation and radiation should be excluded from the preseason estimation. Anyway, the large difference in the preseason estimation is wired, and it would be substantially affect the estimation in

the sensitivity estimation.

*Authors:* We checked our analysis of preseason length for both MODIS and GIMMS and added more discussion about this discrepancy in section 4.2. We calculated correlations between the time of SG and mean temperature in a period proceeding mean SG from 15 to 120 days with an increment of 3 days and identify preseason as the period (15-120) in which mean temperature is best correlated with SG. Although the mean SG patterns look similar for MODIS and GIMMS (Figure 1a and 1b), the difference can be larger than 20 days in some regions (Figure S1b). Therefore, the difference in preseason length for temperature is not only propagated from the differences in mean SG but also the interannual variability of SG. The preseason length inferred from MODIS (41±31days) is very close to the preseason length inferred from GIMMS in an earlier longer period over 1982-2005 (43±30 days). The consistent preseason length inferred from MODIS and GIMMS over two different period, and the stronger MODIS SG-temperature coupling makes it more confident to use MODIS NDVI in the available period and GIMMS NDVI data in the earlier period.

In the temperature/precipitation sensitivity of phenology, only significant relationships were recorded and mapped, but the percentage should also be provided. From the results and figure4, it seems quiet large percentage pixels were removed. if 90% were insignificant and removed, then a mean values across the 10% in T-sensitivity would be nonsense. I would suggest providing all data, both insignificant and sig correlations, and calculating the mean values and providing the percentage of sigs.

*Authors:* We replaced the Figure 4 to show all the pixels where the valid preseason

and preseason climate-SG correlation were calculated. We marked the pixels with 90% confidence level with black dots. The percentages of sensitivity at 90% confidence level are given in section 3.3. About 43% of the land fraction shows significant sensitivity of $D_{SG}^{M}$ to $T_m$ ($p < 0.1$) compared with 13% of the land fraction with significant sensitivity of $D_{SG}^{G}$ to $T_m$. About 11% of the land fraction shows significant sensitivity of $D_{SG}^{M}$ to $P_c$ ($p < 0.1$) as compared with 3% of the land fraction with significant sensitivity of $D_{SG}^{G}$ to $P_c$. We add the mean sensitivity of SG to temperature and precipitation in section 3.3 for all the pixels. For the biome-scale sensitivity of SG to temperature, we found that filtering with ($p<0.1$) criteria tends to exclude pixels with lower sensitivity. But the stronger sensitivity inferred by MODS SG than that inferred by GIMMS SG remains with/without p-value filtering. The biome-scale sensitivity of SG to temperature for MODIS versus GIMMS for all p-value and p-value<0.1 are plotted below (Figure R1).

[Figure]

Figure R1 The biome-scale sensitivity of SG to temperature for MODIS versus GIMMS

3) The reference should be provided in many arguments, such as L37, what's means of several changes? Need references; L114, need references, also why moisture? Generally by T and photoperiod, rather air or soil moisture.

***Authors:*** We rephrased these sentences and references are amended.

We agree with the reviewer that temperature and photoperiod are very important controller of phonological cycle in temperate and boreal vegetation. In the regions with permanent or periodic water stress, water availability mediates the phonological cycle. For example, Delayed spring phenology in response to increased air temperature has been reported in East Asia semiarid regions due to reduced precipitation (Shen et al., 2011; Yu et al., 2003). In a Mediterranean forest and in a mediterranean shrubland, rainfall pattern plays an important role in regulating the phonological change (Peñuelas et al., 2004).

Shen, M., Y. Tang, J. Chen, X. Zhu, Y.  Zheng (2011). Influences of temperature and precipitation before the growing season on spring phenology in grasslands of the central and eastern Qinghai-Tibetan Plateau. Agricultural and Forest Meteorology, 151, 1711–1722. https://doi.org/10.1016/j.agrformet.2011.07.003

Yu, F., Price, K. P., Ellis, J., & Shi, P. (2003). Response of seasonal vegetation development to climatic variations in eastern central Asia. Remote Sensing of Environment, 87, 42–54. https://doi.org/10.1016/S0034-4257(03)00144-5

Peñuelas, J., I. Filella, X. Zhang, L. Llorens, R. Ogaya, F. Lloret, P. Comas, M. Estiarte, J. Terradas (2004). Complex spatiotemporal phonological shifts as a responses to rainfall changes. New Phytologist, 161, 837-846.

---

## Author Comment (AC2) · 15 Oct 2018

**Responses to Anonymous Referee #2**

Summary:

Xu et al. manuscript presents a comparison of spring greenup (SG) and its sensitivity to temperature and precipitation over the northern hemisphere inferred from two remote sensing NDVI products (MODIS and GIMMS) over the period 2001-2013. They aim at exploring the uncertainties in NDVI SG trends and sensitivity to climate. They conclude that both products are consistent in mid latitudes both for SG and its sensitivity to climate, but show different magnitude and trends in high latitudes.

General Comments:

The analysis performed in this study is timely, but the main message and the novelty of the study remain unclear.

First, the authors did not really assess the uncertainties induced by NDVI products and the approach used. Previous studies already highlighted differences in SG and its temporal trend estimated by several approaches and different NDVI products (Chang et al. 2016, Wang et al. 2015, Ding et al. 2015 for example). It is already known that main uncertainties in estimating SG are found in high latitudes. Xu et al. went one step further by comparing SG sensitivity to climate between products but mainly concluded about observed differences, not the uncertainty behind, which in the end led to the same conclusion than previous studies. Because differences in SG estimates leads to differences in pre-season length, the authors compared sensitivities that are not really comparable. The uncertainty in sensitivity to climate results from the propagation of the uncertainty in SG estimates, however these aspects are poorly discussed in the manuscript.

*Authors:* We sincerely thank the reviewer for the constructive remark. In the revision, we have edited and rewritten parts of this manuscript. Further analyses are provided. Our responses and detailed edits are indicated below in the point-to-point responses.

Secondly, the methodology suffers from major flaws. Only one method is used to smooth and fit the data while previous studies highlighted a strong impact of the smoothing method (Atkinson et al. 2012) and approach used to estimate SG. Moreover we don't have information about the performance of the approach. The authors did not take into account partial correlations between temperature and precipitation, thus leading to weak interpretations of the results (Fu et al. 2014). Finally the filtering of data performed in the analysis is sometimes unjustified or incomplete (see specific comments). The authors should refine their research question to be in adequation with their approach or go deeper in the analysis of uncertainty propagation from NDVI products to the estimation of climate sensitivity of SG.

*Authors:* We add a paragraph in section 2.4 to elucidate why we use a single method to smooth NDVI time series and derive greenup-date other than multi-methods. The common used regression methods to reconstruct NDVI time series and derive SG include Savitzky-Golay fitting method, spline smoothing, asymmetric Gaussian functions, double logistic function, and harmonic analysis of times series. These methods are valid in fitting NDVI gaps and reducing noise (Cai et al. 2017), however, they can make differences in estimating phonological stages (Cong et al., 2013). It is hard to justify which method is better (Atkinson et al, 2012). In order to reduce the mixed uncertainty of reconstruction methods and NDVI products, here we used one regression method to reconstruct the NDVI series. The double logistic method uses least-square fitting to half growing season (Zhang et al., 2003). It is more robust than

other methods in estimating the vegetation seasonal dynamics, when there is no local calibration (Cai et al., 2013). Atkinson et al. (2012) also proved that the double logistic method is reliable to smooth the noise, when it is applied to a single growth cycle. Following reviewer's suggestion, we further specified our research objectives in Introduction. We refined our analysis and made more comparison between other recent studies with different products and methods and our results. The uncertainty due to sensors resolution and algorithm are provided in discussions.

Atkinson, P. M., C. Jeganathan,  J. Dash, C. Atzberger (2012). Inter-comparison of four models to smoothing satellite sensor time-series data to estimate vegetation phenology. Remote Sensing of Environment, 123, 400-417.

Cai, Z., P. Jönsson, H. Jin, L. Eklundh (2017). Performance of smoothing method for reconstructing NDVItime-series and estimating vegetation phenology from MODIS data. Remote Sensing, 9, 1271, doi:10.3390/rs9121271.

Cong, N., T. Wang, H. Nan, Y. Ma, X. Wang, R. B. Myneni, S. Piao (2013). Changes in satellite-derived spring vegetation green-up date and its linkage to climate in China from 1982 to 2010: a multi-method analysis. Global Change Biology, 19, 881-891, doi:10.1111/gcb.12077.

Zhang, X., Friedl, M.A., Schaaf, C.B., Strahler, A.H., Hodges, J.C.F., Gao, F., Reed, B.C., Huete, A. (2003). Monitoring vegetation phenology using MODIS. Remote Sensing of Environment, 84, 471–475.

**Specific Comments:**

L.96: the paper from cong et al. is not an evidence that methods in estimating SG has no impact on resulting trends and sensitivity. Especially for sensitivity to climate that

requires the estimation of the pre-season length, there will be a propagation of errors that will influence final results.

*Authors:* Thanks for this insightful suggestion. We rephrased this sentence and in response to reviewer's general comment above, we add a paragraph in section 2.4 to provide more information about the methods to smooth NDVI time series and derive greenup-date. Cong et al. (2013) proposed that the different methods can lead to varied magnitude of SG shift, however, the signs of SG trend are consistent across regions and vegetation types over the same period, when the methods are applied to the same NDVI products. We use the same method and aim to prove that the conflicted SG shift is caused by the different NDVI products and the different NDVI based SG shift propagate the uncertainties in determining the SG sensitivity to climate changes.

L.135: now CRU-NCEP v.8 is extended to 2017. Remove "recently extended"

*Authors:* We rephrased this sentence.

L.145: maybe use the median value.

*Authors:* It would be interesting to evaluate the difference between the mean value and median value when resampling the NDVI products from a high resolution to a low resolution. The commonly used resampling methods include averaging, bicubic, bilinear interpolation and nearest neighbor. Here we keep the spatial averaging method that has been widely applied to NDVI resampling by other studies, e.g. Zeng et al. (2013), Fensholt et al. (2006) and Busetto et al. 2008.

Zeng, F.-W. , Collatz, J. G., Pinzo, J. E., Ivanoff, A. (2013) Evaluating and quantifying the climate-driven interannual variability in Global Inventory Modeling

and Mapping Studies (GIMMS) Normalized Difference Vegetation Index (NDVI3g) at global scales, Remote Sensing, 5, 3918-3950.

Fensholt, R., T.T.Nielsen, S. Stisen (2006) Evaluation of AVHRR PAL and GIMMS 10-day composite NDVI time series products using SPOT-4 vegetation data for the African continent, International Journal of Remote Sensing, 2006, 2719-2733.

Busetto, L.,M. Michele,R.Colombo(2008)Combining medium and coarse spatial resolution satellite data to improve the estimation of sub-pixel NDVI time series, Remote Sensing of Environment, 112, 118-131.

L.142 & 179: Is 15days observation fine enough to estimate sensitivity to climate change properly? Moreover, is it significant to estimate the pre-season length with a 3days step when observations are performed every 15 days?

*Authors:* NDVI time series at 15-day scale is too coarse to indicate the phonological stages. So that, we fit the NDVI time series to a finer scale to obtain the vegetation growth trend and seasonality. Here, we fitted NDVI to daily scale to match the climate data. The double logistic method allows for fitting and smoothing GIMMS NDVI at 15-day (Sobrino and Julien, 2011) or MODIS NDVI at 16-day scale (Hird and McDermid, 2009). As NDVI is fitted and smoothed to a daily scale, it is finer enough to match the pre-season with 3-day step.

Hird, J. N., G. J. McDermid (2009). Noise reduction of NDVI time series: An empirical comparison of selected techniques. Remote Sensing of Environment, 113, 248-258.

Sobrino, J. A. and Y. Julien (2011). Global trends in NDVI-derived parameters obtained from GIMMS data. International Journal of Remote Sensing, 32,

4267-4279.

L.179-183: explain why?

*Authors:* The temporal and spatial distribution of precipitation is heterogeneous. Therefore, we use the relative variation in precipitation to take the variation and baseline precipitation in each pixel.

L.184: the authors should use partial correlations to take into account co-variations of climate.

Authors: Following the reviewer's suggestion, we further analyzed the partial correlation between SG and preseason temperature (Figure R2) and precipitation (Figure R3). We found that the pattern of the partial correlation is close to our calculated Pearson Correlation for both temperature and precipitation. Using the partial correlation, we can reach the same conclusion that temperature overwhelms precipitation as the major driver of the spring phenology.

[Figure]

Figure R2 Partial correlation (left panel) and Pearson correlation (right panel)

between SG and preseason temperature for GIMMS (a, b) and MODIS (c, d).

[Figure]

Figure R3 Partial correlation (left panel) and Pearson correlation (right panel) between SG and preseason precipitation for GIMMS (a, b) and MODIS (c, d).

L.187 & 195: why removing positive correlations? Several studies highlighted different

behaviours according to species and regions (Zhang et al. 2016 for example). By removing positive correlations you start the analysis by assuming that vegetation respond all the time negatively to climate change which is not true. If the aim is to compare both products the authors should keep all the information available.

Authors: In our revision, we supplemented the pixels with insignificant response to temperature and precipitation (Figure 4). As we focused on the green-up and climate control in spring, it is reasonable excluded the positive interannual correlation between preseason climate and green-up date shift. In Zhang et al. (2016), the observations indicated the spring temperature warming induced the advancement of leaf unfolding in 99.7% of the studied cases, either significant (48.8%) or insignificant (50.9%). Only 0.3% of the studied cases showed spring warming and delayed leaf unfolding relationships.

L.194: OLS or SMA regression. In this case SMA regression are more appropriate

*Authors:* The Ordinary Least Square (OLS) regression is appropriate for the relationship between two variables that are clarified with one independent variable and one dependent variable. The Standard Major Axis (SMA) regression is suitable for the cases in which the independent and dependent variables are not clear. In our study, the phonological change is a response to climate drivers, i.e. SG is the dependent variable and climate drivers are independent variables.

L.195: why not excluding non-vegetated pixels? It would improve the analysis.

*Authors:* we screened the pixels with maximum NDVI < 0.1 that potentially excluded the non-vegetated pixels.

L.195: to avoid a bias due to the number of significant pixels, the authors should compare only pixels for which significant sensitivities can be estimated for both products.

*Authors:* In our revision, we displayed the pixels with insignificant response to temperature and precipitation and marked the pixels with 90% confidence level with black dots (Figure 4). Our results are based on the pixels with significant sensitivity. But in response to reviewer #1, we add the mean sensitivity of SG to temperature and precipitation in section 3.3 for all the pixels. For the biome-scale sensitivity of SG to temperature, we found that filtering with (p<0.1) criteria tends to exclude pixels with lower sensitivity. But the stronger sensitivity inferred by MODS SG than that inferred by GIMMS SG remains with/without p-value filtering.

L243: check partial correlations between SG temperature and precipitation.

*Authors:* Please see our responses above for L.184.

L247: a significant correlation does not mean a control. Please reformulate

*Authors:* We accept reviewer's suggestion and rephrased.

L.252-256: how does it relate to changes in SG?

*Authors:* The Pearson Correlation Coefficient is calculated between time of SG and preseason temperature and precipitation (Section 2.4). The higher PCC indicates a

better correlation between time SG and preseason climate.

L.271: Does +/- 7 or 4mm means a significant change in precipitation? .

*Authors:* In response to this reviewer comment, we provide a figure below to show the total preseason precipitation (mm) correlated to GIMMS derived SG. The distribution of preseason precipitation is very heterogeneous. But the maximum 7mm yr$^{-1}$ and -4mm yr$^{-1}$ in preseason means a strong change in most regions. Due to the strong heterogeneity, we use the relative change of precipitation to assess the sensitivity of spring phenology to precipitation change.

[Figure]

Figure R4 Total preseason precipitation correlated to GIMMS derived SG.

L.275: because we don't have the same pre-season length it is difficult to conclude.

*Authors:* The preseason is determined as the period in which mean temperature (or precipitation) is best correlated with SG. The difference in preseason length indicated the uncertainties in SG-climate links propagated from uncertainties in SG predictions. If we use the same pre-season, e.g. spring season, the sensitivity of SG to climate would be calculated by regression of SG derived from different NDVI products with the same climate, from which there must be different sensitivities of SG to climate due to different SGs. But the SG-climate links would be concealed. In reality, the period during which the temperature controls the interannual variability and long term trend of spring phenology is debated depending on different methods and varied

across regions and biomes. The Europe-wide earlier growing season was analyzed in correlation with the warming spring during February-April (Chmielewski and Rotzer, 2001) while variation in spring phenology of temperate tree species in south-west France was attributed to the temperature variation during March-May (Vitasse et al., 2009). The correlation analysis between the spring phenology and temperature in different period imply that temperatures in different time-scales have been reported to play different roles across regions and biomes. In horticultural woody perennials in northeastern USA, the warming trend in annual temperature is well correlated to the spring advance, while the warming trend in monthly March and April temperature was not very significant (Wolfe et al., 2005). The phenology changes in some Mediterranean species were most correlated to the temperature changes in the months preceding the phenological events while other species correlated to the annual temperature changes (Peñuelas et al., 2002). In 254 records from nine European coutries, 19% of the phenophases are highest correlated with the temperature in the month of onset, 63% with preceding month and 18% with 2 months earlier (Menzel et al., 2006).

Chmielewski, F. M., and T. Rotzer, 2001: Response of tree phenology to climate change across Europe. Agricultural and Forest Meteorology, 108, 101–112.

Vitasse, Y., A. J. Porté, A. Kremer, R. Michalet, S. Delzon (2009) Response of canopy duration to temperature changes in four temperate tree species: relative contributions of spring and autumn leaf phenology, Oecologia, 161, 187-198.

Wolfe, D. W., M. D. Schwartz, A. N. Lakso, Y. Otsuki, R. M. Pool, N. J. Shaulis (2005) Climate change and shifts in spring phenology of three horticultural woody perennials in northeastern USA, International Journal of Biometeorology, 49, 303-309.

Peñuelas, J., I. Filella, P. E. Comas (2002), Changed plant and animal life cycles from 1952 to 2000 in the Mediterranean region, Global Change Biology, 8, 531-544.

Menzel, A., T.H. Aparks, N. Estrella, E. Koch, A. Aasa et al. (2006) Erupoean pheanological response to climate change matches the warming pattern, Global Change Biology, 12, 1969-1976.

L.280: why it is not responsive to precipitation?

*Authors:* Temperature has long been recognized as the dominant factor that alters SG. For example, the records of tree SG over 100 years from England (Thompson & Clark, 2008) and flowering in the northeastern U.S. (Miller-Rushing & Primack, 2008) have chronicled advances of 3–8 days for each 1°C increase in air temperature over the 1 or 2 months preceding the SG or flowering. European larch in northern Italy Alpine regions has advanced at a rate of 7 days per °C increase in spring air temperature (Busetto et al., 2010). The vegetation types with earlier mean SG in lower latitude are more sensitive to temperature increases and show larger advances over the historical period (Shen et al., 2014), and 88% of the latitudinal variability in the SG trend can be explained by preseason temperature (Shen et al., 2015). Even in the regions with permanent or periodic water stress, water availability is recognized as a secondary factor that mediates the phonological cycle (Seghieri et al., 2012).

Thompson, R., & Clark, R. M. (2008). Is spring starting earlier? Holocene, 18, 95–104. https://doi.org/10.1177/0959683607085599

Miller-Rushing, A. J., & Primack, R. B. (2008). Global warming and flowering times in Thoreau's concord: a community perspective. Ecology, 89, 332–341. https://doi.org/10.1890/07-0068.1

Busetto, L., Colombo, R., Migliavacca, M., Cremonese, E., Meroni, M.,

Galvagno, M., Pari, E. (2010). Remote sensing of larch phonological cycle and

analysis of relationships with climate in the Alpine region. Global Change Biology,

16, 2504–2517. https://doi.org/10.1111/j.1365-2486.2010.02189.x

Shen, M., Tang, Y., Chen, J., Yang, X., Wang, C., Cui, X., … Cong, N. (2014).

Earlier-Season Vegetation Has Greater Temperature Sensitivity of Spring Phenology

in Northern Hemisphere. PLoS ONE, 9(2), e88178.

https://doi.org/10.1371/journal.pone.0088178

Seghieri, J., Carreau, J., Boulain, N., De Rosnay, P., Arjounin, M., & Timouk,

F. (2012). Is water availability really the main environmental factor controlling the

phenology of woody vegetation in the central Sahel? Plant Ecology, 213, 861–870.

Shen, M., Cong, N., & Cao, R. (2015). Temperature sensitivity as an

explanation of the latitudinal pattern of green-up date trend in northern Hemisphere

vegetation during 1982–2008. International Journal of Climatology, 35, 3707–3712.

https://doi.org/10.1002/joc.4227

L.283: percent compared to which value?

*Authors:* We rephrased this sentence.

L.300: interesting result. It is consistent with field observations over Europe (Fu et al.

2015)

*Authors:* Thanks. This is an interesting study and we cited it.

L.312: recent studies showed that CCI is better than NDVI in detecting phonological

changes for evergreen (Gamon et al. 2016). That may explain the behaviour of

evergreen vegetation in this study.

*Authors:* Thank you for suggesting this publication. We add Gamon et al. 2016 in our discussion.

Technical comments:

Figures 2abc are not cited in the text

*Authors:* Figure 2a, b and c are now cited in section 3.1.

As you compare both products figure s1 is more relevant than figure 1. Try to use absolute or relative comparisons in the main figures and put absolute values in supp, also for fig3. Moreover the scale make it difficult to see where differences are null.

*Authors:* We changed the Figure 1 and Figure 3 as suggested.

Atkinson, Peter M., et al. "Inter-comparison of four models for smoothing satellite sensor time-series data to estimate vegetation phenology." Remote sensing of environment
123 (2012): 400-417.

Wang C, Cao R, Chen J, Rao Y, Tang Y. 2015. Temperature sensitivity of spring vegetation phenology correlates to within-spring warming speed over the Northern Hemisphere. Ecol Indic. 50:62–68.

Ding M, Li L, Zhang Y, Sun X, Liu L, Gao J,Wang Z, Li Y. 2015. Start of vegetation growing season on the Tibetan Plateau inferred from multiple methods based on GIMMS and SPOT NDVI data. J Geog Sci. 25:131–148. Better consistency with observations is the threshold

Chang, Qing & Zhang, Jiahua & Wenzhe, Jiao & Yao, Fengmei. (2016). A

comparative analysis of the NDVIg and NDVI3g in monitoring vegetation phenology

changes in the Northern Hemisphere. Geocarto International. 33. 1-20.

10.1080/10106049.2016.1222633.

Zhang, H.,W. Yuan, S. Liu, andW. Dong. 2015. Divergent responses of leaf

phenology to changing temperature among plant species and geographical regions.

Ecosphere

6(12): 250. http://dx.doi.org/10.1890/ES15-00223.1

Gamon, John A., et al. "A remotely sensed pigment index reveals photosynthetic

phenology in evergreen conifers." Proceedings of the National Academy of Sciences

113.46 (2016): 13087-13092.

---

## Author Comment (AC3) · 15 Oct 2018

**Responses to Anonymous Referee #3**

Summary

The manuscript compares the spring greenup date (SG) obtained using the method of Zhang et al. (2003) applied to the GIMMS dataset and the MODIS MOD13C1 dataset. It also compares the trend of the chronological regression on the same period, and the sensitivity of the SG to the climate before the SG. The results are completed by the analysis of the trend before the launch of MODIS.

General evaluation

The current manuscript has a nice first objective, which is to explore the impact of the input remote sensing dataset to which the SG algorithm is applied. However it requires more analysis to achieve this objective. If the objective is to evaluate the impact of the choice of the input dataset then the causes of the differences should be the target. This would allow increasing the confidence in the trend analysis based on the two datasets. The suggestion on the role of the transition from AVHRR2 to AVHRR3 should also be explored in depth, as this is quite an interesting opening. In the following I suggest some changes and some previous articles that should be considered. The second objective on the sensitivity of SG to climate cannot be achieved without improvements on the work on the SG determination, as there is strong uncertainty on the SG.

*Authors:* We sincerely thank the reviewer for the valuable comments and suggestions. Our responses are indicated below in the point-to-point responses.

Major comments

1/ The results report differences in the SG obtained with MODIS and with AVHRR and in the trend. The manuscript suggests possible explanations but does not attempt

to determine the reasons clearly. In-depth exploration of the causes of these differences should be carried out as it is the aim of the manuscript. Possible sources of uncertainties, including spatial resolution changes with incidence angle, preprocessing, processing are explored in :

Helman, D. (2018) Land surface phenology: What do we really "see" from space? Science of The Total Environment, 618, 665 – 673

*Authors:* Thank you for suggesting this timely published work. Helman (2018) provides comprehensive information about the uncertainties and limitations in determining phenology from satellite data. In our revision, we add discussions about the difference that can be brought from different sensors, e.g. the NDVI by MODIS and GIMMS were retrieved from a different spatial resolution. The retrieved NDVI is a mixture of different vegetation species with diverse phenologies, bare soil and even water bodies dependent on the spatial resolution (Helman, 2018). The different resolution lead to the NDVI difference and NDVI difference propagates biases to SGs.

Moreover, a key issue is the snowmelt. It is well-known that the detected SG is related to snowmelt and not to vegetation if the snow is not correctly treated (Moulin et al. 1997, Shabanov et al. 2002). Solutions have been proposed (Suzuki et al. 2003, Delbart et al. 2005, 2006, Thomson et al. 2015, Jin et al. 2017 for examples). In the current manuscript the snow issue is treated differently between the two dataset preprocessing, thus the results differ. It is necessary to assess the uncertainty of the two SG datasets through a comparison to ground observations of leaf expansion, in order to analyse the impact of the snow rejection methods.

Delbart, N., Kergoat, L., Le Toan, T., Lhermitte, J., & Picard, G. (2005) Determination of

phenological dates in boreal regions using normalized difference water index. Remote Sensing of Environment, 97, 26–38.

Delbart, N., Le Toan, T., Kergoat, L., & Fedotova, V. (2006) Remote sensing of spring phenology in boreal regions: A free of snow-effect method using NOAA-AVHRR and SPOT-VGT data (1982-2004). Remote Sensing of Environment, 101, 52–62.

Jin, H., Jönsson, A.M., Bolmgren, K., Langvall, O., & Eklundh, L. (2017) Disentangling remotely-sensed plant phenology and snow seasonality at northern Europe using MODIS and the plant phenology index. Remote Sensing of Environment, 198, 203–212.

Moulin, S., Kergoat, L., Viovy, N., & Dedieu, G. (1997) Global-scale assessment of vegetation phenology using NOAA/AVHRR satellite measurements. Journal of Climate, 10, 1154–1170.

Shabanov, N.V., Zhou, L., Knyazikhin, Y., Myneni, R.B., & Tucker, C.J. (2002) Analysis of interannual changes in northern vegetation activity observed in AVHRR data from 1981 to 1994. IEEE Transactions on Geoscience and Remote Sensing, 40, 115–130.

Suzuki, R., Nomaki, T., & Yasunari, T. (2003) West-east contrast of phenology and climate in northern Asia revealed using a remotely sensed vegetation index. International Journal of Biometeorology, 47, 126–138.

Thompson, B.G. (2015) Using phase-spaces to characterize land surface phenology in a seasonally snow-covered landscape. Remote Sensing of Environment, 166, 178–190.

*Authors:* We agree with the reviewer that snow cover is a big challenge in determining the SG in the high latitudes. We enhanced our analysis of uncertainties in

snow effect on phonological determination and the options to improve phenology estimation from alternative indices. The snow influences phenology determination in two ways. On the one hand, the snow cover led to NDVI gaps during the dormancy season. As a result, the time series of NDVI cannot be adequately fitted during the transitional snow melting and vegetation greening season (Zhou et al., 2015). On the other hand, the overlapped time of snowmelt and greenup leads complexity in greenup determination. Spring greenup date is almost the same as snow-end date some high latitude regions (Zeng and Jia, 2013). Therefore, in high latitudes with seasonal snowpack, the beginning of the growing season is often determined by snowmelt rather than temperature (Semenchuk et al., 2016). The normalized difference water index method (Delbart et al., 2004; Delbart et al., 2006), plant phenology index method (Jin et al. 2017), normalized difference vegetation index-normalized difference infrared index phase-space method (Thompson et al., 2015) are alternatives to improve the NDVI-based phonological metrics. We are conducting continued studies focusing on the complexity of snowmelt and phenology determination for specific boreal regions.

Delbart, N., L. Kergoat, T. Le Toan, J. Lhermitte, G. Picard (2005). Determination of phenological dates in boreal regions using normalized difference water index. Remote Sensing of Environment, 97, 26–38.

Delbart, N.,T. Le Toan, L. Kergoat, V. Fedotova (2006). Remote sensing of spring phenology in boreal regions: A free of snow-effect method using NOAA-AVHRR and SPOT-VGT data (1982-2004). Remote Sensing of Environment, 101, 52–62.

Jin, H., A. M. Jönsson, K. Bolmgren, O. Langvall, L. Eklundh (2017). Disentangling remotely-sensed plant phenology and snow seasonality at northern

Europe using MODIS and the plant phenology index. Remote Sensing of Environment, 198, 203–212.

Semenchuk, P. R., M. A. K. Gillespie, S. B. Rumpf, N. Baggesen, B. Elberling, E. J. Cooper (2016). High Arctic plant phenology is determined by snowmelt patterns but duration of phonological periods is fixed: an example of periodicity. Environmental Research Letters, 125006. DOI: 10.1088/1748-9326/11/12/125006.

Thompson, B.G. (2015). Using phase-spaces to characterize land surface phenology in a seasonally snow-covered landscape. Remote Sensing of Environment, 166, 178-190.

Zeng, H. and G. Jia (2013). Impacts of snow cover on vegetation phenology in the Arctic from satellite data. Advances in Atmospheric Sciences, 30, 1421-1432.

Zhou, J., L. Jia, M. Menenti (2015). Reconstruction of global MODIS NDVI time series: Performance of Harmonic ANalysis of Time Series (HANTS). Remote Sensing of Environment, 163, 217-228. DOI: 10.1016/j.rse.2015.03.018

2/ The pre-season length differs when so the variations of the climate variables differ. Thus the trend of the preseason climate from the two SG datasets should not be compared.

*Authors:* In the revision, we focus more on the relation between SG and preseason climate (Figure 3) to indicate the climate-SG links. We kept our analysis of length and climate trend in preseason because the difference in the preseason is propagated from the conflicts of SG estimation in GIMMS and MODIS. Although the length of preseason length differed when inferred by GIMMS and MODIS SGs, the pattern of climate trend in preseason is very close. This consistent preseason climate pattern, however, did not lead to a consistent SG response between GIMMS and MODIS. The

difference in vegetation dynamics leads to uncertainties in understanding climate-vegetation couplings.

3/ The trends are reported if the p-value is less than 0.1. This is not a good value as it is too high. Maximum is generally 0.05, which is very high already.

*Authors:* To address the reviewer's concern, we revised the trend at 95% confidence level in section 3.1. At a 95% confidence level, the trends are slightly different from the trend at a 90% confidence level. The sign of trend remains.

5/ The GIMMS dataset and the MOD13C1 dataset are composite products with a compositing period of 15 and 16 days. This has a strong impact on SG uncertainty. This is why PAL product should be prefered (10 day composite) of 8-day MODIS dataset. The effect of the compositing period duration must be explored.

*Authors:* In the discussion, we added discussion about the uncertainty that may be raised by the different composite technique. Both GIMMS NDVI3g and MOD13C1 were generated using daily surface reflectance product to a similar composite interval. However, the MODIS applied the constrained-view angle-maximum value composite while GIMMS applied maximum value composite. The maximum value composite cannot completely remove atmospheric effect (Pinzo and Tucker 2014) and the different composite technique can cause the value difference in the same interval (van Leeuwen et al., 1999; Gallo et al., 2004).

Gallo, K. P., L. Ji, B. Reed, J. Dwyer, J. Eidenshink (2004). Comparison of MODIS and AVHRR 16-day normalized difference vegetation index composite data. Geophysical Research Letters, 31, L07502, doi:10.1029/2003GL019385.

Van Leeuwen, W. J. D., A. R. Huete, T. W. Laing (1999). MODIS vegetation index compositing approach: a prototype with AVHRR data. Remote Sensing of Environment, 69, 264-280.

Pinzo, J. E. and C. J. Tucker (2014). A Non-stationary 1981-2012 AVHRR NDVI3g time series. Remote Sensing, 6, 6929-6960.

6/ Reported trends should be compared to those from :

Gonsamo, A. & Chen, J.M. (2016) Circumpolar vegetation dynamics product for global change study. Remote Sensing of Environment, 182, 13–26.

Park, T., Ganguly, S., Tømmervik, H., Euskirchen, E.S., Høgda, K.-A., Karlsen, S.R., Brovkin, V., Nemani, R.R., & Myneni, R.B. (2016) Changes in growing season duration

and productivity of northern vegetation inferred from long-term remote sensing data. Environmental Research Letters, 11, 084001.

*Authors:* Thank you for suggesting these two studies on phenology in Circumpolar region. We made comparison with our results in our revision. SPOT VGT phenology products showed continuously advanced SG trend over 1999-2013, in consistent with MODIS. But the magnitude and spatial distribution differs (Gonsamo and Chen, 2016). The results by GIMMS NDVI are comparable with our results in Northern high latitude, advanced SG trend before 2000 and delayed trend thereafter (Park et al., 2016).

Minor comments

1/ Analysing the sensitivity of SG to temperature through linear correlation is not totally

convincing. The phenology models are well known to be unlinear (non linear effects are mentionned in the discussion) and parameterized with thresholds. See for example :Hänninen, H. (1990) Modelling bud dormancy release in trees from cool and temperate regions. Acta Forestalia Fennica, 213, 1-47.The consequence is that, for example in an arctic ecosystem, a warming from -15 C to -5C in March will have no impact on SG whereas a warming from 2C to 3C in may would have a strong impact. Thus changes in sensitivity of SG to temperature changes are expected.

*Authors:* In Hänninen (1990), five types of phonological models with combined chilling and thermal forcings. The interaction between chilling and thermal requirements leads to a non-linear response of budburst to temperature change. In the northern hemisphere north of 40°N, the chilling requirement are always fulfilled (Zhang et al., 2007). A pure thermal forcing model showed a better spring phenology prediction than the combined chilling and thermal forcing model (Yang et al., 2012). The manipulated experiments proved that the temperate trees are linearly correlated with the spring warming (Fu et al., 2012). For example, The cubic function and the linear model predicted a similar leaf unfolding rate based on hourly average temperatures recorded in a Florida commercial greenhouse during two times of the year. The linear relationship is reliable in most observations (86%) at 3657 stations in 22 European countries for linking spring phenology and temperature (Jochner et al., 2016).

Fu, Y. H., M. Campioli, G. Deckmyn, I. A. Janssens (2012). The impact of winter and spring temperatures on temperate trees budburst dates: results from an Experimental climate manipulation. PLoS One, 7, e47324.

Yang, X., J. F. Mustard, J. Tang, H. Xu (2012). Regional-scale phenology modeling based on meteorological records and remote sensing observations. Journal

of Geophysical Research: Biogeosciences, 117,

https://doi.org/10.1029/2012JG001977

Zhang, X., D. Tarpley, and J. Sullivan (2007), Diverse responses of vegetation

phenology to a warming climate, Geophys. Res. Lett., 34, L19405,

doi:10.1029/2007GL031447.

Jochner, S., T. H. Sparks, J. Laube, A. Menzel (2016). Can we detect a

nonlinear response to temperature in European plant phenology? International Journal

of Biometeorology, 60, 1551-1561.

2/ The style of the writing is often hard to read and the text should be clarified.

*Authors:* We adjust some paragraphs of methods and results and polished the writing.

---

## Author Comment (AC4) · 15 Oct 2018

[revised manuscript text omitted]

**2. Data and Method**

**2.1 Study area and biomes**

We restricted our analysis to north of 30°N, since that is the region where temperate and boreal vegetation dominates and phenology is expected to be most strongly controlled by the annual cycle of temperature (Linderholm, 2006; Fu et al.

2014; Shen et al., 2015; Güsewell et al., 2017), and regulated by water availability (Peñuelas et al., 2004; Shen et al., 2011) and photoperiod (Way and Montgomery,

2015; Singh et al., 2017). In order to analyze the phenology and its response to climate across biomes, we used global mosaics of collection 6 MODIS data products (MCD12Q1) in the IGBP classification of land cover types with spatial resolution of

0.5° x 0.5° to mask the satellite-based SG results. The global mosaics of MCD12Q1

with geographic coordinates of latitude and longitude on the WGS 1984 coordinate reference system (EPSG: 4326) (Channan et al., 2014) were re-projected from standard MCD12Q1 with 500m resolutions (Friedl et al., 2010). We used the IGBP

land cover classification for 9 biomes in 2012 (Table S1): Evergreen Needleleaf

Forest (ENF), Deciduous Needleleaf Forest (DNF), Deciduous Broadleaf forest (DBF), Mixed Forest (MF), Open Shrublands (OS), Woody Savannas (WS),

Grassland (GL), Permanent Wetland (PW), and Cropland (CP). We distinguish the grassland to the north of 60°N (GLN), which is more likely to be tundra, from grassland in the temperate south (GLS) due to their expected differences in climate and its controls on phenology.

**2.2 Climate reanalysis**

We calculated daily mean air temperature ($T_m$) and cumulative precipitation ($P_c$) from 6-hourly, half-degree resolution CRU-NCEP (Climate Research Unit-

National Centers for Environmental Prediction) v6 reanalysis to identify the preseason climate associated with SG. The CRU-NCEP v6 dataset extended to 2014, is a combination of CRU TS v3.2 0.5° x 0.5° monthly climatology and NCEP reanalysis

2.5° x 2.5° with six hours time step available in near real time (http://forge.ipsl.jussieu.fr/orchidee/wiki/Documentation/Forcings).

**2.3 NDVI products**

We used the latest version NDVI time series (GIMMS NDVI3g) derived from the AVHRR instrument on board the NOAA satellite series. This dataset spans the period from July 1981 to December 2013 with spatial resolution of 1/12° and bimonthly temporal resolution (Pinzon and Tucker, 2014).

We also used the 16-day MODIS NDVI composites (MOD13C1, collection 6)

at 0.05° spatial resolution, and further performed data quality control. We regridded both GIMMS and MODIS NDVI data to 0.5° x 0.5° resolution by taking the mean value in a 0.5° x 0.5° pixel to match the spatial resolution of the CRU-NCEP

reanalysis. We screened the pixels with annual maximum NDVI <0 to exclude the non-vegetated pixels. For GIMMS NDVI3g, the algorithm has improved snow-melt detection and the pixels recognized with snow or ice were filled with average seasonal profile or spline interpolation (Pinzon and Tucker, 2014).  The pixels flagged with snow/ice were given the NDVI values with the values from the previous nearest period without snow influence. Even though, the filled values are very close to zero in the dormant season and the near-zero values are smoothed by the double logistic method or piecewise logistic method described in section 2.3. SGs were derived from

GIMMS NDVI 2001-2013 to fit the time period of MOD13C1 NDVI product.

**2.4 Determination of SG and preseason climate**

We determined the preseason duration following the method of Shen et al.

(2014), but with a different climate reanalysis product and a different method for calculating SG. The common used regression methods to reconstruct NDVI time series and derive SG include Savitzky-Golay fitting method, spline smoothing, asymmetric Gaussian functions, double logistic function, and harmonic analysis of times series. These methods are valid in fitting NDVI gaps and reducing noise (Cai et al. 2017), however, can make differences in estimating phonological stages (Cong et al., 2013).   In order to reduce the mixed uncertainty of reconstruction methods and

NDVI products, here we used one regression method to reconstruct the NDVI series.

The double logistic method uses least-square fitting to half growing season (Zhang et al., 2003). It is more robust than other methods in reducing noise (Hird and

McDermid, 2009) and estimating the vegetation seasonal dynamics, when there is no local calibration (Cai et al., 2013). As we applied the double logistic method to a single growth cycle, it is reliable to smooth noise (Atkinson et al., 2012).

[revised manuscript text omitted]

The lagged snow phenology advancement implies that snow complication in determine SG in the cold regions is still present at a warmer climate. To reduce the snow effect on spring phenology determination, the normalized difference water index method (Delbart et al., 2004; Delbart et al., 2006), plant phenology index method (Jin et al. 2017), normalized difference vegetation index- normalized difference infrared index phase-space method (Thompson et al., 2015) are alternatives to improve the NDVI-based phonological metrics.

**4.2 SG dates sensitivities to climate**

The SG to preseason climate sensitivity by MODIS and GIMMS showed varied degree of vegetation-climate seasonal coupling. The differences in MODIS and GIMMS SG propagate the conflicts to the preseason length. However, the $L_{PT}^M$ is very close to $L_{PT}^G$ ($=43\pm30$days) in an earlier longer period over 1982-2005 (Xu et al., 2018). The higher correlation between MODIS SG and preseason temperature indicates stronger MODIS SG-climate relationships. The consistent preseason length inferred from MODIS over 2001-2013 and GIMMS over 1982-2005, and stronger MODIS SG-temperature coupling indicate more reliable MODIS NDVI in the available period and GIMMS NDVI data in the earlier period. The stronger MODIS NDVI to temperature correlation than GIMMS NDVI was also reported in central Europe, where the correlation between temperature and August NDVI anomalies were analyzed (Kern et al., 2016). The stronger SG-temperature coupling than precipitation is consistent with our previous study of SG to climate sensitivity over 1982-2005 (Xu et al., 2018). MODIS inferred stronger SG-temperature sensitivity in the northern boreal and Arctic biomes can be explained by the site-level observation that temperature sensitivity of phenology is greater in colder, higher latitude sites than in warmer regions (Prevéy et al., 2017). At the colder sites, the small changes in temperature may constitute greater relative changes in thermal budget (Oberbauer et al., 2013), so that the warming impacts on vegetation are amplified. This explanation is not applicable to the GIMMS NDVI inferred SG response to temperature that vegetation with earlier growing season is more sensitive to temperature (Shen et al., 2014).

The sensitivity of GIMMS SG to temperature increased over 2001-2013 in relative to that over 1988-2000. Our results showed SG to temperature sensitivity increased most significantly in Arctic grassland (44.6%), followed by other boreal biomes (open shrubland (41.2%), permanent wetland (35.9%), woody savanna (31.1%)

and deciduous needleleaf forest (17.6%)). The magnitudes of enhanced sensitivity are even larger when we compare 2001-2013 SG-temperature sensitivity with a longer period over 1982-2005 (Xu et al., 2018). Compare with the period 1982-2005, SG- temperature sensitivity of the northern biomes (deciduous needleleaf forest, woody savanna, open shrublands and permanent wetlands) all increased more than 50% over

2001-2013 with stable inter-biome sensitivity variation $(r = 0.91, p < 0.01)$.

The increased sensitivity of SG to temperature for boreal biomes has not been well investigated. In the contrary, temperature sensitivity of spring greenup may decline under warmer climate because (1) insufficient winter chilling may delay the spring greenup in spite of continued spring warming (Yu et al., 2010), (2) when spring greenup starts earlier, shorter photoperiod can limit the potential of leaf development (Chmielewski & Götz, 2016), (3) greenup may respond nonlinearly to temperature and be saturated at a high temperature (Caffarra & Donnelly, 2011), and (4) under warmer condition, the preseason duration of thermal forcing can be reduced, which declines the SG-temperature sensitivity (Güsewell et al., 2017). The vegetation growth (represented by NDVI) to temperature sensitivity was reported declining in the growing season (April-October) based on GIMMS NDVI over 1982-2012 linked to water stress (Piao et al., 2014). In temperate ecosystems, the lower NDVI to temperature sensitivity coincidently occurred with increased drought events.  While in the arctic ecosystem, the lowered sensitivity of NDVI to temperature may be explained by increases in heat waves because the physiological response of photosynthesis to temperature is nonlinear with lower sensitivity under warmer conditions (Piao et al., 2014). The higher interannual temperature variability can also cause higher variations in water supply, thus the declined coupling between vegetation growth and interannual variability of growing season temperature, generally in semiarid regions (Wu et al., 2017). The wetting preseason in mid to east of the United States, Western Canada, Northern land along Norway and Northwestern

Russia may partly enhanced SG-temperature if the enhancement is validated.

**4.3 Uncertainties in SG as derived by MODIS and GIMMS NDVI**

With SG as inferred using GIMMS over the period 1988-2000 and as inferred using MODIS over 2001-2013, we found that the trend is advanced continuously in response to a continuing trend in preseason warming. The uncertainties in the SG

trend and its climatic sensitivity arise when SG as inferred using GIMMS, MODIS, and other sensors and in situ observations are compared together over a similar period after 2000, during which the main conflicts in SG trend were found. Our results coincide with other studies that GIMMS NDVI inferred an opposite trend of SG

before and after 2000 in the circumpolar Arctic (Park et al., 2016). SPOT VGT

retrieved a continuously advanced SG trend over 1999-2013 in the circumpolar region (>45 °N), in consistent with MODIS SG, although the magnitude and spatial distribution of the advancement are different between SPOT and MODIS (Gonsamo and Chen, 2016). Wang et al. (2016) and Zhang et al. (2013) proposed that quality issues may present in GIMMS NDVI, which can bias vegetation growth sensitivity and growth trend. Instead of using continuous GIMMS SG over 1982-2011, Zhang et al. (2013) merged datasets of GIMMS SG over 1982-2000 and SPOT-VGT SG over

2001-2011 to detect SG trend due to data quality issues with GIMMS NDVI in most parts of western Tibetan Plateau, according to the findings of opposite GIMMS SG

trend to SPOT-VGT and MODIS SG trend over the period 2001-2006. With this merged data record, the SG trend continuously advanced in Tibetan Plateau over

1982-2011. This result is consistent with the SG trend derived from tree-ring data (Yang et al., 2017). On the contrary, continuous GIMMS SG over 1982-2006 inferred delayed SG trend after mid-1990s over Tibetan Plateau (Yu et al., 2010). At the North

Hemisphere scale, GIMMS SG (1982-2008) showed significant decadal variation and declining SG shift: advanced 5.2 days over 1982-1999, but only advanced 0.2 days over 2000-2008 (Jeong et al., 2011). However, the merged GIMMS (1982-2006) and

MODIS (2002-2012) showed SG shift over 2002-2012 (-6 days decade$^{-1}$) is about three times larger than that over 1982-2002 (-2 days decade$^{-1}$), which is interpreted as enhanced SG advancement and its response to temperature over time (Wang et al.,

2016). For the varied timing of SG derived from different products, Zhang et al. (2017)

suggested intersensor calibrations to reduce the difference between vegetation index products and exclusion of the low quality phonology timing. The ground observations are solutions to validate the remote sensed phenology. However, in situ observations and remote sensed phenology differed no matter how accurate they are retrieved (Gonsamo and Chen, 2016), due to the scale and resolution issues.

These SG shift uncertainties after 2000 are more likely to be explained by the differences in the NDVI products that implied the opposite SG trend, anomalies north of 50ºN and biome-scale SG-temperature sensitivities. The spectrum range difference of MODIS and AVHRR sensor channels is a main contribute to the NDVI differences.

MODIS NDVI is derived from bands 1(620-670nm) and 2 (841-876nm) of the

MODIS on board NASA's Terra satellite whereas GIMMS NDVI is derived from bands 1(580-680nm) and 2 (725-1100nm) of AVHRR. Furthermore, the NDVI by

MODIS and GIMMS were retrieved from a different spatial resolution. The retrieved

NDVI is a mixture of different vegetation species with diverse phenologies, bare soil and even water bodies dependent on the spatial resolution (Helman, 2018). Both

GIMMS NDVI3g and MOD13C1 were generated using daily surface reflectance product to a similar composite interval. However, the MODIS applied the constrained-view angle- maximum value composite while GIMMS applied maximum value composite. The maximum value composite cannot completely remove atmospheric effect (Pinzo and

Tucker 2014) and the different composite technique can cause the value difference in the same interval (Gallo et al., 2004).

[revised manuscript text omitted]

Delbart, N., L. Kergoat, T. Le Toan, J. Lhermitte, G. Picard (2005). Determination of phenological dates in boreal regions using normalized difference water index. Remote Sensing of Environment, 97, 26–38.

Delbart, N.,T. Le Toan, L. Kergoat, V. Fedotova (2006). Remote sensing of spring phenology in boreal regions: A free of snow-effect method using NOAA-AVHRR and SPOT-VGT data (1982-2004). Remote Sensing of Environment, 101, 52–62.

Durpaire, J. P., T. Gentet, T. Phulpin, and M. Arnaud (1995). "Spot-4 Vegetation Instrument: Vegetation Monitoring on a Global Scale." Acta Astronautica, 35, 453-459.

Fensholt, R. and  I.  Sandholt (2005). Evaluation of the MODIS and NOAA AVHRR vegetation indices with in situ measurements in a semi-arid environment. International Journal of Remote Sensing, 26, 2561-2594.

Fensholt, R., T.T. Nielsen, S. Stisen (2006). Evaluation of AVHRR PAL and
GIMMS 10-day composite NDVI time series products using SPOT-4 vegetation data
for the African continent. International Journal of Remote Sensing, 27, 13, 2719-2733,
DOI:10.1080/01431160600567761

Fensholt, R., K. Rasmussen, T. T. Nielsen, C. Mbow (2009). Evaluation of earth
observation based long term vegetation trends-intercomparing NDVI time series trend
analysis consistency of Sahel from AVHRR GIMMS, Terra MODIS and SPOT VGT
data. Remote Sensing of Environment, 113, 1886-1898.

Fensholt, R. and S. R. Proud (2012). Evaluation of Earth Observation based
global long term vegetation trends- Comparing GIMMS and MODIS global NDVI
time series. Remoste Sensing of Environment, 119, 131-147.

Friedl, M.A., D. Sulla-Menashe, B. Tan, A. Schneider, N. Ramankutty, A.
Sibley and X. Huang (2010). MODIS Collection 5 global land cover: Algorithm
refinements and characterization of new datasets, 2001-2012, Collection 5.1 IGBP
Land Cover, Boston University, Boston, MA, USA.

Fu, Y. H., M. Campioli, Y. Vitasse, H. J. De Boeck, J. Van den Berge, H.
AbdElgawad, H. Asard, S. Piao, G. Deckmyn, I. A. Janssens (2014). Variation in leaf
flushing date influences autumnal senescence and next year's flushing date in two
temperate tree species. Proceedings of the National Academy of Sciences of the
United States of America, 111, 7355-7360. https://doi.org/10.1073/pnas.1321727111

Fu, Y. H., S. Piao, M. Op de Beeck, N. Cong, H. Zhao, Y. Zhang, A. Menzel, I.
A. Janssens (2015). Recent spring phenology shifts in western Central Europe based
on multiscale observations. Global Ecology and Biogeography, 23, 1255-1263.

Gallo, K. P., L. Ji, B. Reed, J. Dwyer, J. Eidenshink (2004). Comparison of
MODIS and AVHRR 16-day normalized difference vegetation index composite data.
Geophysical Research Letters, 31, L07502, doi:10.1029/2003GL019385.

Gamon, J. A., K. F. Huemmrich, C. Y. S. Wong, I. Ensminger, S. Garrity, D. Y.
Hollinger, A. Noormets, J. Peñuelas (2016). A remotely sensed pigment index reveals
photosynthetic phenology in envergreen conifers. Proceedings of the National
Academy of Sciences of the United States of America, 113, 13087-13092.

Gonsamo, A. and J. M. Chen (2016) Circumpolar vegetation dynamics
product for global change study. Remote Sensing of Environment, 182, 13-26.

Güsewell, S. R. Furrer, R. Gehrig, B. Pietragalla (2017). Changes in
temperature sensitivity of spring phenology with recent climate warming in
Switzerland are related to shifts of the preseason. Global Change Biology, 23, 5189-
5202.

Helman, D. (2018). Land surface phenology: what do we really 'see' from space?
Science of The Total Environmenta, 618, 665-673.

Hird, J. N., G. J. McDermid (2009). Noise reduction of NDVI time series: An
empirical comparison of selected techniques. Remote Sensing of Environment, 113,
248-258.

Jeong, S.-J., C.-H. Ho, H.-J. Gim, M. E. Brown (2011). Phenology shifts at start
vs. end of growing season in temperate vegetation over the Northern Hemisphere for
the period 1982-2008. Global Change Biology, 17, 2385-2399, doi: 10.1111/j.1365-
2486.2011.02397.x

Jin, H., A. M. Jönsson, K. Bolmgren, O. Langvall, L. Eklundh (2017). Disentangling remotely-sensed plant phenology and snow seasonality at northern Europe using MODIS and the plant phenology index. Remote Sensing of Environment, 198, 203–212.

Kern, A., H. Marjanović, Z. Barcza. (2016). Evaluation of the quality of NDVI3g Dataset against collection 6 MODIS NDVI in central Europe between 2000 and 2013. Remote Sensing, 8, 955; doi:10.3390/rs8110955.

Linderholm, H. W. (2006). Growing season changes in the last century. Agricultural and Forest Meteorology, 137, 1-14.

Marshall, M., E. Okuto, Y. Kang, E. Opiyo, M. Ahmed. (2016). Global assessment of vegetation index and Phenology Lab (VIP) and Global Inventory Modeling and Mapping Studies (GIMMS) version 3 products. Biogeosciences, 13, 625-639.

Moulin, S., L. Kergoat, N. Viovy, G. Dedieu (1997). Global-scale assessment of vegetation phenology using NOAA/AVHRR satellite measurements. Journal of Climate, 10, 1154-1170.

Myneni, R. B., C. D. Keeling, C. J. Tucker, G. Asrar, R. R. Nemani (1997). Increased plant growth in the northern high latitudes from 1981 to 1991. Nature, 386, 698-702.

Oberbauer, S.F., S. C. Elmendorf, T. G. Troxler et al. (2013). Phenological response of tundra plants to background climate variation tested using the International Tundra Experiment. Philosophical Transactions of the Royal Society B: Biological Sciences, 368, 20120481. DOI: 10.1098/rstb.2012.0481

Park, T., S. Ganguly, H. Tømmervik, E. S. Euskirchen, K.-A. Høgda, S. R. Karlsen, V. Brovkin, R. R. Nemani, R. B. Myneni (2016). Changes in growing season duration and productivity of northern vegetation inferred from long-term remote sensing data. Environmental Research Letters, 11. 0.1088/1748-9326/11/8/084001

Peñuelas, J., I. Filella, X. Zhang, L. Llorens, R. Ogaya, F. Lloret, P. Comas, M. Estiarte, J. Terradas (2004). Complex spatiotemporal phonological shifts as a responses to rainfall changes. New Phytologist, 161, 837-846.

Piao, S., H. Nan, C. Huntingford, P. Ciais, P. Friedlingstein, S. Sitch, S. Peng, A. Ahlshtröm, J. G. Canadell, N. Cong, S. Levis, P. E. Levy, L. Liu, M. R. Lomas, J. Mao, R. B. Myneni, P. Peylin, B. Poulter, X. Shi, G. Yin, N. Viory, T. Wang, X. Wang, S. Zaehle, N. Zeng, Z. Zeng, A. Chen (2014). Evidence for a weakening relationship between interannual temperature variability and northern vegetation activity. Nature Communications, doi:10.1038/ncomms6018.

Pinzon, J. E., C. J. Tucker (2014). A non-stationary 1981-2012 AVHRR NDVI3g time series. Remote Sensing, 6, 6929-6960. doi:10.3390/rs6086929

Prevéy, J., M. Vellend, N. Rüger, R. D. Hollister, A. D. Bjorkman, I. H. Myers-Smith, S. C. Elmendorf, K. Clark, E. J. Cooper, B. Elberling, A. M. Fosaa, G. H. R. Henry, T. T. Høye, I. S. Jónsdóttir, K. Klanderud, E. Lévesque, M. Mauritz, U. Molau, S. M. Natali, S. F. Oberbauer, Z. A. Panchen, E. Post, S. B. Rumpf, N. M. Schmidt, E. A. G. Schuur, P. R. Semenchuk, T. Troxler, J. M. Welker, C. Rixen (2017). Greater temperature sensitivity of plant phenology at colder sites: implications for convergence across northern latitudes. Global Change Biology, 23, 2660-2671, DOI: 10.1111/gcb.13619.

Shen, M., Y. Tang, J. Chen, X. Zhu, Y. Zheng (2011). Influences of temperature and precipitation before the growing season on spring phenology in grasslands of the central and eastern Qinghai-Tibetan Plateau. Agricultural and Forest Meteorology, 151, 1711–1722. https://doi.org/10.1016/j.agrformet.2011.07.003

Shen M., Y. Tang Y, J. Chen, X. Yang, C. Wang, X. Cui, Y. Yang, L. Han, L. Li, J. Du, G. Zhang, N. C (2014). Earlier-Season Vegetation Has Greater Temperature Sensitivity of Spring Phenology in Northern Hemisphere. PLoS ONE 9(2): e88178. DOI:10.1371/journal.pone.0088178

Shen, M., N. Cong, R. Cao (2015). Temperature sensitivity as an explanation of the latitudinal pattern of green-up date trend in northern Hemisphere vegetation during 1982–2008. International Journal of Climatology, 35, 3707–3712. https://doi.org/10.1002/joc.422

Semenchuk, P. R., M. A. K. Gillespie, S. B. Rumpf, N. Baggesen, B. Elberling, E. J. Cooper (2016). High Arctic plant phenology is determined by snowmelt patterns but duration of phonological periods is fixed: an example of periodicity. Environmental Research Letters, 125006. DOI: 10.1088/1748-9326/11/12/125006.

Singh, R. K., T. Svystun, B. AlDahmash, A. M. Jönsson, R. P. Bhalerao (2017). Photoperiod- and temperature-mediated control of phenology in trees – a molecular perspective. New Phytologist, 213, 511-524.

Thompson, B.G. (2015). Using phase-spaces to characterize land surface phenology in a seasonally snow-covered landscape. Remote Sensing of Environment, 166, 178-190.

[revised manuscript text omitted]

Zhou, L., C. J. Tucker, R. K. Kaufmann, D. Slayback, N. V. Shabanov, R. B. Myneni (2001) Variations in northern vegetation activity inferred from satellite data of vegetation index during 1981-1999. Journal of Geophysical Research: Atmosphere, 106, 20069-20083. Doi: 10.1029/2000JD000115